# Satellite-observed relationships between land cover, burned area and atmospheric composition over the southern Amazon.

Emma Sands[1], Richard Pope[2,3], Ruth M. Doherty[1], Fiona M. O'Connor[4,5], Chris Wilson[2,3], and Hugh Pumphrey[1]

[1]School of Geosciences, University of Edinburgh, Edinburgh, United Kingdom
[2]School of Earth and Environment, University of Leeds, Leeds, United Kingdom
[3]National Centre for Earth Observation, University of Leeds, Leeds, United Kingdom
[4]Met Office, Exeter, United Kingdom
[5]Department of Mathematics and Statistics, Global Systems Institute, University of Exeter, Exeter, United Kingdom

**Correspondence:** Emma Sands (e.g.sands@ed.ac.uk)

**Abstract.** Land surface changes can have substantial impacts on biosphere-atmosphere interactions. In South America, rainforests abundantly emit biogenic volatile organic compounds (BVOCs), which coupled with pyrogenic emissions from deforestation fires, can have substantial impacts on regional air quality. We use novel and long-term satellite records of five trace gases: isoprene ($C_5H_8$), formaldehyde (HCHO), methanol ($CH_3OH$), carbon monoxide (CO), and nitrogen dioxide ($NO_2$); aerosol optical depth (AOD); vegetation (land cover and leaf area index); and burned area. We characterise the impacts of biogenic and pyrogenic emissions on atmospheric composition for the period 2001 to 2019 in the southern Amazon, a region of substantial deforestation. The seasonal cycle for all of the atmospheric constituents peaks in the dry season (August-October) and year-to-year variability in CO, HCHO, $NO_2$, and AOD is strongly linked to burned area. We find a robust relationship between broadleaf forest cover and total column $C_5H_8$ ($R^2 = 0.59$), while burned area exhibits an approximate 5th root power law relationship with tropospheric column $NO_2$ ($R^2 = 0.32$), both in the dry season. Vegetation and burned area together show a relationship with HCHO ($R^2 = 0.23$). Wet season AOD and CO follow the forest cover distribution. The land surface variables are very weakly correlated with $CH_3OH$, suggesting other factors drive its spatial distribution. Overall, we provide a detailed observational quantification of biospheric process influences on southern Amazon regional atmospheric composition, which in future studies can be used to help constrain the underpinning processes in Earth System Models.

## 1 Introduction

Ten million hectares of forest on average were cut down globally each year over 2010-2020 (Ritchie and Roser, 2021). Such land cover changes can substantially modify the emissions of biogenic gases and aerosols, for example biogenic volatile organic compounds (BVOCs) (Fowler et al., 2009; Pacifico et al., 2012). BVOCs are emitted during photosynthesis and particular plant development stages, e.g. leaf maturation, flowering or senescence, or as a response to stresses on plants, such as droughts and insect infestations (Loreto and Fares, 2013). Estimates of the global emission of isoprene ($C_5H_8$), the globally dominant

BVOC, are within the ranges of 300-600 $TgC\ yr^{-1}$ (Arneth et al., 2011; Cao et al., 2021; Szopa et al., 2021). This large range is predominantly driven by uncertainties in the emission rates from different plant functional types (Szopa et al., 2021).

While BVOCs are associated with a wide range of vegetation, particular plant species or functional types emit different amounts of specific BVOCs.This has driven the use of emission factors, empirically-derived values used to scale calculated trace gas emissions, to describe the sensitivity of emission strength to plant type (e.g. Guenther et al. (1995, 2012); Pacifico et al. (2011); Weber et al. (2023)). For example, isoprene is associated with tropical broadleaf trees, while monoterpenes ($C_{10}H_{16}$) are primarily associated with needleleaf trees (Artaxo et al., 2022), which has implications for the geographical distribution of different BVOC emissions. In the Amazon rainforest, isoprene and methanol ($CH_3OH$), both analysed in this study, are the most strongly emitted biogenic compounds based on mixing ratio measurements (Yáñez-Serrano et al., 2020). Although biogenic emissions are the largest sources of isoprene, monoterpenes and methanol, they are also emitted during biomass burning (Akagi et al., 2011; Ciccioli et al., 2014; Bates et al., 2021).

Satellite measurements of formaldehyde (HCHO), a common oxidation product of BVOCs, are often used to estimate BVOC emissions (e.g. Palmer et al. (2006); Millet et al. (2008); Marais et al. (2012); Kefauver et al. (2014); Stavrakou et al. (2015); Strada et al. (2023)). However, the pyrogenic source of HCHO is more significant than the pyrogenic emission of isoprene. Further, HCHO is an oxidation product of many other non-biogenic gases, introducing challenges to the interpretation of the data (Freitas and Fornaro, 2022; Palmer et al., 2007). Recent work has enabled the measurement of isoprene column densities from space (Fu et al., 2019; Wells et al., 2020, 2022; Palmer et al., 2022). The new isoprene measurements have created opportunities to address the uncertainties in regional isoprene emissions. This is particularly relevant for regions experiencing land cover change and in the southern hemisphere, where ground level measurements are sparse, despite the major BVOC emission sources being located in the tropical southern hemisphere (Paton-Walsh et al., 2022). In this study, we analyse these satellite-derived datasets of column isoprene alongside the more established HCHO product to quantify vegetation-driven changes in composition in the southern Amazon (see section 2.1).

The Amazon and neighbouring savannas and grasslands experience significant impacts from fire activity, which have been reviewed by Pivello (2011). Fires in the region have both natural and anthropogenic causes. Lightning can ignite the savanna and grassland vegetation; these ecosystems are fire-dependent, meaning many of the species have adapted to recurrent fires. However, unlike the savanna region, the Amazon rainforest is sensitive to burning and the ecosystem can be destroyed through fire activity. In the 21st century, the majority of wildfires in Brazil have anthropogenic causes, as natural vegetation (e.g. the rainforest) is removed for agriculture. In fact, fire is one of the major causes of land cover change globally (Heald and Spracklen, 2015). This suggests that regions of land cover change driven shifts in biogenic emissions will also often experience substantial pyrogenic impacts on the atmospheric composition.

Nitrogen dioxide ($NO_2$) is a trace gas measurable from space that is primarily emitted during combustion, whether that is biomass burning or anthropogenic emissions. Due to its short lifetime, $NO_2$ can be indicative of biomass burning activity. In the context of land cover change through biomass burning, the burning of different land cover types is associated with different $NO_2$ emission rates (Schreier et al., 2014). Wiedinmyer et al. (2023) assign the highest $NO_2$ emission factors for the burning of tropical forests, followed by savanna grasslands, and lower values for crops and woody savanna. This is in contrast to earlier

emission inventory estimates from Akagi et al. (2011), where the biomass burning emission factor for $NO_x$ for tropical forests was lower than for savannas, although uncertainties were substantial. These biomass burning $NO_x$ emissions can play a key role in enabling ozone ($O_3$) formation from VOC emissions, including BVOCs (e.g. Helas et al. (1995)). Particularly in the Amazon, a $NO_x$-limited region abundant in BVOCs, $O_3$ production increases substantially over the rainforest in plumes of anthropogenic or pyrogenic emissions (Kuhn et al., 2010; Bela et al., 2015).

Fires also emit significant amounts of aerosols and carbon monoxide (CO) (Badr and Probert, 1994; Wiedinmyer et al., 2023) and are a key factor in driving regional aerosol concentrations in the dry season (Reddington et al., 2015). In addition to the pyrogenic sources of these species, in areas such as the Amazon forest, there may be substantial biogenic contributions, for example, through the role of BVOCs in the formation and growth of secondary organic aerosol (SOA) (Artaxo et al., 2013; Shrivastava et al., 2017; Artaxo et al., 2022). SOA, as a component of particulate matter (PM) air pollution, is detrimental to human health (Kim et al., 2015). In the Amazon, the biogenic source of aerosols and CO has a greater relative contribution to regional atmospheric composition outside the wildfire season, when pyrogenic emissions decrease (Artaxo et al., 2022). Understanding the drivers of biogenic and pyrogenic emissions of $O_3$ and PM precursors is important for both regional air quality and climate.

Overall, land cover change influences regional atmospheric composition through biogenic and pyrogenic sources, which can vary substantially over time and space. In this study, we aim to quantify the relevance of vegetation and fire to spatial and temporal variations in regional atmospheric composition over the last two decades, as observed using satellite remote sensing, over an area of significant land cover change and biomass burning: the southern Amazon.

Six measures of trace gases and aerosols in the atmosphere were chosen to represent a range of chemical species that may be impacted by biogenic and/or pyrogenic emissions. These are isoprene, methanol, formaldehyde, carbon monoxide, nitrogen dioxide and aerosol optical depth (AOD), which indicates the amount of aerosol in the atmospheric column. These will be referred to as atmospheric constituents throughout this paper. We compare this range of atmospheric constituents to vegetation and fire proxies using both new and complementary satellite datasets to build a comprehensive picture of the relative impact of both biogenic and pyrogenic sources on regional atmospheric composition during the period 2001-2019. The paper will introduce the region, data and methodology in section 2, before looking at the spatial and seasonal distribution of the atmospheric constituents in the region and their links to both land cover and burned area (section 3 for results and section 4 for discussion and conclusions).

## 2    Data and Methodology

### 2.1    Study region

The southern Amazon is one of the regions that has undergone substantial deforestation in the 21[st] century. The region investigated in this study: 50°- 70°W, 5°- 25°S (Fig. 1); covers the majority of the "arc of deforestation", which forms the epicenter of Amazon deforestation (Santos et al., 2021; Silva Junior et al., 2021; Reddington et al., 2015). The area includes parts of the southern Amazon, as well savannas and grasslands such as the Cerrado and Sarmiento.

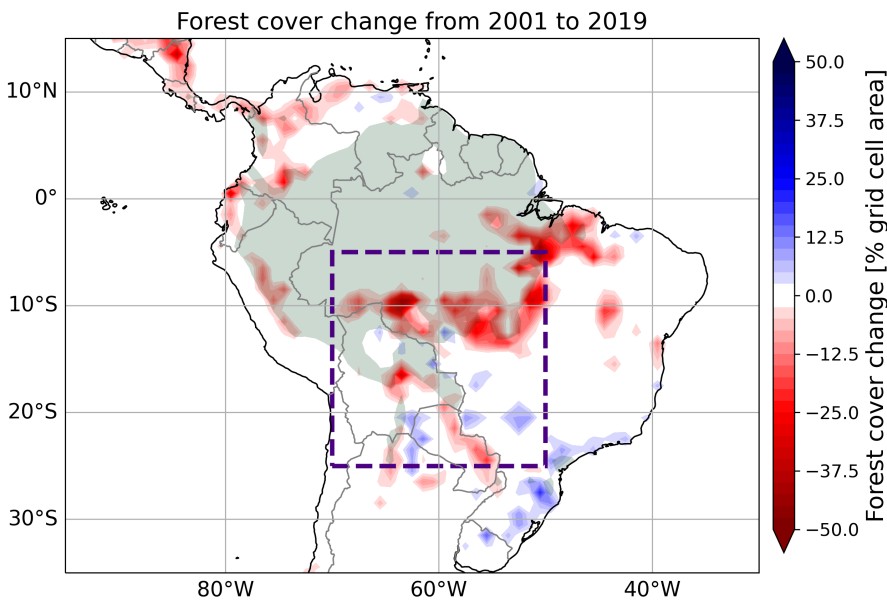

**Figure 1.** Change in forest cover (% of grid cell area deforested or re/afforested) between 2001 and 2019 calculated using MODIS land cover data. The region analysed in this study is marked with the purple box.

## 2.2 Remote sensing data sources

### 2.2.1 Land surface data

The Moderate Resolution Imaging Spectroradiometer (MODIS) land cover data at 0.05° resolution (product MCD12C1, Friedl and Sulla-Menashe (2015), last access July 2022) was chosen as a measure of vegetation type based on its long temporal record of annual data (Table 1), and a comparison of the magnitude and timing of Amazon deforestation in three land cover datasets (see Supplement). The MODIS sensor operates on two satellites: Terra (launched in December 1999) and Aqua (launched in May 2002). For 2001-2020, the time period for which the data was used, the Terra and Aqua crossing times were, respectively, 10:30 and 13:30 local time. Both had sun synchronous, near polar orbits and an altitude of 705 km. The land cover data is available since 2001 and combines retrievals from both satellites. From the multiple land cover classifications available in the MCD12C1 product, the 17-class International Geosphere-Biosphere Programme classification (IGBP, Loveland and Belward (1997)) was chosen for this project, as it has been used in relevant work, such as the FINNv2.5 fire emission inventory (Wiedinmyer et al., 2023). The C6 Modis land cover product is assessed to have an accuracy of 73.6% (Sulla-Menashe et al., 2019).

The leaf area index (LAI) data is also a MODIS product (MOD15A2H, Myneni et al. (2021), last access January 2024) and can represent vegetation abundance. The LAI product is based on measurements from the Terra satellite at an 8-day temporal

**Table 1.** Summary of satellite datasets analysed in the study. All datasets were analysed on $1° \times 1°$ spatial resolution. Lifetimes are taken from Jacob (1999); Holloway et al. (2000); Pacifico et al. (2009); Hodzic et al. (2016); Wells et al. (2020); Bates et al. (2021); Pommier (2023)

| Variable | Source | Temporal Period | Period of Data | Lifetime |
|---|---|---|---|---|
| Land Cover | MODIS, Terra and Aqua | Annual | 2001-2020 | NA |
| Leaf Area Index (LAI) | MODIS, Terra | Monthly/Annual | 2001-2020 | NA |
| Burned area | GFED4.1s | Monthly | 2001-2016 | NA |
| Isoprene | CrIS, Suomi-NPP | Monthly | 2012-2020 | <1 day |
| Methanol | IASI, Metop-A/B | Monthly | 2008-2018 | Days-Months |
| Formaldehyde (HCHO) | OMI, EOS Aura | Monthly | 2005-2018 | <1 day |
| Carbon Monoxide (CO) | Mopitt, Terra | Monthly | 2001-2019 | Months |
| Nitrogen Dioxide ($NO_2$) | OMI, EOS Aura | Monthly | 2005-2020 | <1 day |
| Aerosol Optical Depth (AOD) | MODIS, Terra | Monthly | 2000-2019 | SOA: Days-Weeks |

resolution and, validated against ground measurements, has a root mean square error of 0.69 across all biomes (Devadiga and Nickeson, 2023). It is available since February 2000 to present-day. We used the LAI product to calculate the annual and monthly mean values for the period 2001-2020 at $1°$ by $1°$ resolution in Google Earth Engine (Gorelick et al., 2017).

The burned area to represent fire activity was obtained from Version 4.1s of the Global Fire Emissions Database (GFED4.1s, Giglio et al. (2013), last access August 2022). Monthly burned area is available at $0.25°$ spatial resolution from August 2000 to December 2016. The data for 2001-2016 was used in the analysis. For the period of interest the GFED4.1s burned area is predominantly based on the MCD64A1 product, which has been found to have a 68% burned area omission error (Padilla et al., 2015). However, GFED4.1s includes the addition of small fire burned areas, which likely counter some of the omission error, as the GFED4.1s burned area is 37% greater than that of GFED3, which did not include small fire estimates (van der Werf et al., 2017). A summary of the land surface and atmospheric composition datasets is provided in Table 1.

### 2.2.2 Atmospheric composition data

Gridded total isoprene columns for 2012-2020 at $0.5° \times 0.625°$ spatial resolution at monthly resolution were obtained from Wells et al. (2020). These data are derived from measurements taken using the Cross-track Infrared Sounder (CrIS), a Fourier-transform spectrometer, onboard the Suomi-NPP satellite. Suomi-NPP was launched in October 2011 and has a sun-synchronous orbit and near-global twice daily coverage, with a daytime local overpass time of 13:30. Isoprene column densities were calculated using two isoprene infrared absorption features in the spectral range 890-910 $cm^{-1}$. This novel data product has been quality assessed through comparison against ground-based isoprene column measurements in the Amazon, which found the retrieved isoprene column amounts differ by 20% to 50% compared to ground-based column measurements (Fu et al., 2019; Wells et al., 2020, 2022).

The total column methanol data, another recently developed dataset, comes from the Infrared Atmospheric Sounding Inter-ferometers (IASI) on board the MetOp-A and MetOp-B satellites. These Eumetsat MetOp satellites have/had (Metop-A was

125 deorbited in 2021) sun-synchronous polar orbits at an altitude of around 817 km and local overpass times of 9:30 and 21:30. The daytime (9:30) data for 2008-2018 was produced using the Infrared-Microwave-Sounding (IMS) scheme developed by the Rutherford Appleton Laboratory (RAL) (Pope et al., 2021). Pope et al. (2021) found a systematic difference of around 30% compared to the Atmospheric Tomography Mission (ATom) flight measurements in areas of methanol enhancement, as well as an uncertainty of 40% to 50% for an individual sounding, which will have been reduced here by averaging.

HCHO and $NO_2$ are measured using the Ozone Monitoring Instrument (OMI) located on the Earth Observing System (EOS) Aura satellite. OMI employs spectrometers in visible and ultraviolet wavelengths and provides daily global coverage since late 2004. Aura flies at 705 km and has an equator crossing time of 13:45 local time. Level 2 total column HCHO data was downloaded from the NASA Goddard Earth Sciences Data and Information Services Center (GES DISC, Chance (2007), last access: June 2022). Only pixels with the "good" main quality flag, which includes checks of fit convergence,

column uncertainty and absolute column value, and cloud cover less than 20% were used to calculate monthly mean values. Uncertainties of individual retrievals of the HCHO columns range within 50% to 105%, with HCHO hotspots characterised by lower uncertainty values, and averaging leading to uncertainty reduction (OMI Team, 2012). We found an underlying positive trend in the HCHO data, possibly associated with instrument degradation (Wang et al., 2022), and anomalous values in 2019. Consequently, only the HCHO data for 2005-2018 was used and the monthly values were de-trended based on a remote Pacific

region (see Supplement). This highlights local variations due to biogenic and pyrogenic emissions, as opposed to those driven by changes to the global background.

The tropospheric column $NO_2$ data is the Quality Assurance for Essential Climate Variables (QA4ECV) tropospheric $NO_2$ product (Boersma et al. (2011), last access: December 2022) available as global monthly averages at $0.125° \times 0.125°$ spatial resolution. The data was downloaded for 2005-2020. The monthly mean values only include retrievals with cloud radiance

fractions under 50%, which is approximately equivalent to geometric cloud fractions under 20%. Boersma et al. (2011) esti-mated the uncertainty for individual retrievals of the $NO_2$ tropospheric columns to be 1.0 x $10^{15}$ molecules cm$^{-2}$ + 25% of the retrieval.

AOD is a measure of light attenuation by atmospheric aerosols due to either absorbance or reflectance (Wei et al., 2020). Low AOD values (<0.1) indicate clear sky and low aerosol amounts, while values of 1 suggest very hazy conditions with

150 high aerosol concentrations. Consequently, an increase in aerosol concentration, e.g. due to emissions of particles during combustion, should lead to higher AOD values. AOD is provided as part of a level 3 $1° \times 1°$ spatial resolution monthly product from MODIS measurements on board the Terra satellite (MOD08_M3, Platnick (2015), last access: May 2022). Each monthly statistically derived dataset (SDS) is based on the relevant MODIS Atmosphere Daily Global Joint Product. The quality controlled overland AOD data is available at three wavelengths: 0.47 μm, 0.55 μm and 0.66 μm. AOD retrievals are

155 expected to have errors within $\pm 0.05 + 0.2 \times$ AOD value (Levy et al., 2013; Sayer et al., 2014). The 0.47 μm data for 2000-2019 is used throughout the analysis, as it was found to exhibit the strongest statistical relationship with the land variables (not shown).

The total column CO data is from the Measurement of Pollution in the Troposphere (MOPITT) sensor also on board the Terra satellite. Monthly mean values at $1° \times 1°$ resolution were calculated for 2001-2019 from the version 7 level 2 product (NASA/LARC/SD/ASDC (2000), last access: December 2021). The CO total column values have biases of less than 0.2 x $10^{18}$ molecules cm$^{-2}$ and standard deviations of around 0.2 x $10^{18}$ molecules cm$^{-2}$ compared to NOAA validation sites and a field campaign (Deeter et al., 2017). The data for 2000 was omitted due to large data gaps that year. Only clear-sky scenes are included in the MOPITT retrievals.

## 2.3 Analysis of remote sensing data for the southern Amazon

All observational data was re-gridded to a $1° \times 1°$ horizontal resolution using linear interpolation to provide spatially consistent datasets for analysis and inter-comparison. The atmospheric composition and burned area data were all analysed on a monthly temporal resolution, while the land cover and LAI data were annual, with the exception of calculating the LAI seasonal cycle. To better understand the impacts of seasonality, we separated the monthly data into wet (February, March, April) and dry (August, September, October) seasons. The three month intervals were chosen based on seasonal variations in precipitation and compatibility with previous definitions (Barkley et al., 2009; Reddington et al., 2015). The dry season data was further separated into high burned area (monthly burned areas $\geq 0.04\%$ grid cell area) and low burned area grid cells. The threshold for defining high burned area of $\geq 0.04\%$ was chosen to ensure sufficient sample sizes in each category, while identifying areas with clear fire signals (see Supplement). There is a 6 km difference in elevation within the region with maximum elevations in the south west (SW). As some of the satellite retrievals are associated with high errors over the high altitudes of the Andes, the small portion of the domain >1000 m a.s.l. was not included in our analysis.

Whenever data for different domains is compared (e.g. difference in isoprene over regions of low and high burned area at a given forest cover), the mean across all available data satisfying the domain conditions was calculated. First, the relevant datasets were subset in time so the only years included are those for which land cover, burned area and composition data are all available. Next, the data were split based on the burned area threshold. Each subset (high and low burned area) was then sorted into land cover bins, then the mean and standard error for each bin were calculated.

Regression analysis was used to quantify the relationship between surface variables: land cover, LAI and burned area, and the atmospheric constituents. Spearman rank correlation (Dodge, 2008):

$$r = 1 - \frac{6 \sum_{i=1}^{n} d_i^2}{n(n^2 - 1)}, \tag{1}$$

where r is the Spearman rank correlation coefficient, n is the sample size and $d_i$ represents the difference between the ranks of the $i^{th}$ values from each sample, and ordinary least squared (OLS) and Theil-Sen regression methods (Fernandes and Leblanc, 2005) were utilised. The OLS regression aims to minimise the sum of the squared differences between the observations and the model (residual sum of squares) when fitting a linear model. The Theil-Sen regression is more robust to outliers than OLS regression, as it fits a slope and intercept based on the spatial median of these parameters calculated on subpopulations of the data. Where a clear relationship (OLS coefficient of determination ($R^2$) $\geq 0.25$) was identified between an atmospheric constituent and land cover or burned area, the data were binned by the explanatory variable, and a weighted least squares

(WLS) regression was used to account for the variance of the data in each bin. Throughout the text we use standard errors to represent uncertainty ranges.

## 3    Results

### 3.1    Change in atmospheric constituents, land cover and burned area through time

Over the early 21$^{st}$ century (2001-2019), forest cover averaged over the southern Amazon region decreases by 3.8% from 52.0% to 48.2% of the study region (Fig. 2a). The deforestation is greatest in the north (N) of the domain at around 10° S (see Fig. 1). Generally across the region, forest cover reduces substantially from 2001 to 2013 with a smaller decline thereafter . Over the same period, savanna and grassland expand from 46.7% to 50.5% (Fig. 2b) (see Supplement for separated savanna and grassland time series). The two land cover categories: broadleaf forest and savanna/grassland, display opposite trends through

time (Fig. 2a,b), reflecting that broadleaf forest cover is being replaced by the savanna/grassland modal land cover type.

Mean LAI and total annual burned area averaged over the study region do not have consistent trends over 2001-2019 and 2001-2016, respectively (Fig. 2c,d). Average annual LAI values in the region fluctuate around a value of 3.1 with a standard deviation of 0.06. As the regional estimate of LAI is dependent on all vegetation types in the domain, any decrease in LAI due to decreases in broadleaf forest (high LAI values) may be of a lesser magnitude than year-to-year variability in the vegetation

overall due to, e.g., weather or disease. Burned area also exhibits substantial year-to-year variability, with maximum values observed in 2007 and 2010, while the least burning occurs in 2009. Since 2011, the monthly average burned area has remained below 200 km$^2$ in the peak burning months and the inter-annual variability has decreased.

The year-to-year variability for annual regional-mean CO, NO$_2$ and AOD is similar to that for burned area (minima in 2009 and maxima in 2007 and 2010; Fig. 2h,i,j). These atmospheric constituents also show reduced year-to-year variability from

2011 onwards, with the exception of elevated values in 2015 for AOD and CO. The 2007 and 2010 maxima are also observed in the HCHO record, although the year-to-year variability does not reduce for this trace gas (Fig. 2g). The similarities in the temporal record suggest burned area has an influence on the domain averaged AOD, CO, NO$_2$ and HCHO.

Methanol and isoprene are also characterised by substantial inter-annual variability and neither shows a clear trend through time for the region average (Fig. 2e,f). Consequently, other annually varying factors affecting these constituents, such as

meteorology or changes in other sources and sinks (Pacifico et al., 2009; Wohlfahrt et al., 2015), may have a greater impact than the decrease in broadleaf forest cover, which is limited in its spatial extent, over this time period. It is noted that the annual mean isoprene and methanol datasets do not extend as far back in time as the land cover dataset and the change in broadleaf forest cover is smaller during the period covered by these datasets.

### 3.2    Seasonal cycle in atmospheric composition and burned area

The regional average seasonal cycles of the six atmospheric constituents are found to be similar with a peak in the dry season (August to October) for all species (Fig. 3). Uniquely, isoprene shows a secondary peak earlier in the calendar year in

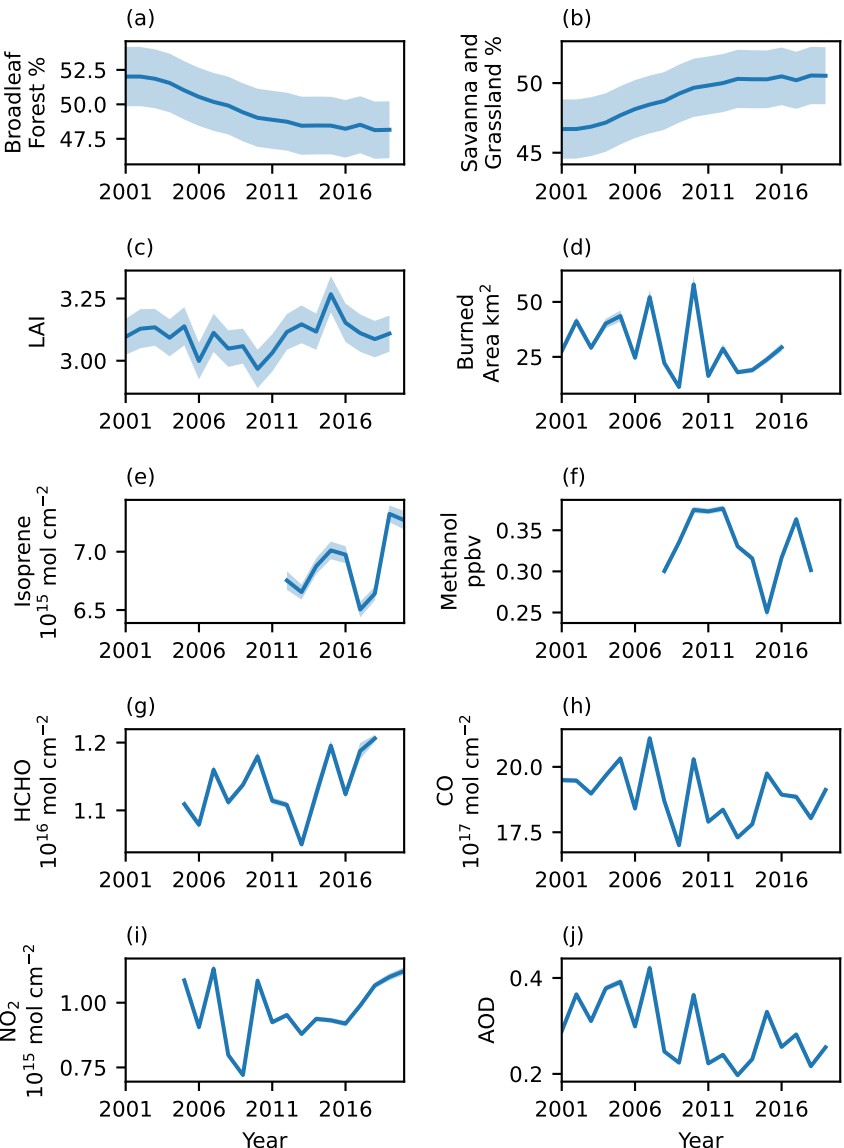

**Figure 2.** Annual mean values for (a) broadleaf forest cover percentage, (b) savanna and grassland cover percentage, (c) LAI, (d) burned area (based on monthly sums) and the atmospheric constituents: (e) isoprene, (f) methanol, (g) HCHO, (h) CO, (i) $NO_2$ and (j) AOD; averaged for the southern Amazon region for all years available for each variable. The shading represents the standard error based on all data included in calculating the annual mean: the values of each grid cell for a given year for the land cover and LAI, or the monthly values of each grid cell for burned area and the atmospheric constituents. The larger amount of data used in calculating the mean values for subplots d-j results in smaller standard errors.

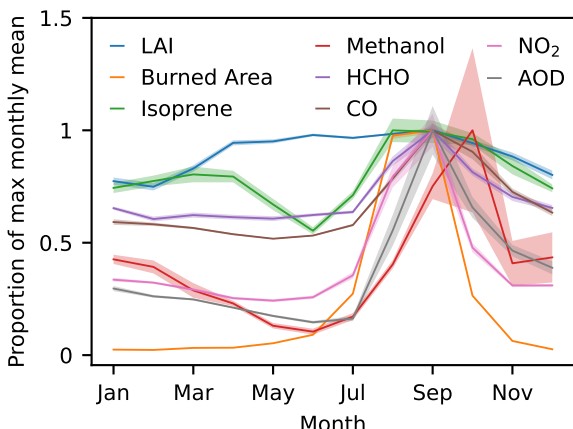

**Figure 3.** Seasonal cycles of domain-averaged LAI, burned area, isoprene, methanol, HCHO, CO, NO$_2$ and AOD for the periods where each dataset is available between 2001-2020. All variables have been normalised (to a value of 1) against their respective regional maximum monthly mean. The shaded areas represent the standard error for each variable for a given month.

March/April, when the isoprene column densities reach 80% of the annual maximum (August regional mean of $8.6 \pm 0.5 \times 10^{15}$ molecules cm$^{-2}$).

Several of the constituents also have co-occurring minima. Isoprene, methanol and AOD all have the lowest values of $4.7 \pm 0.1 \times 10^{15}$ molecules cm$^{-2}$, $0.1 \pm 0.01$ ppbv, and $0.11 \pm 0.004$, respectively, around June. HCHO, CO and NO$_2$ remain more stable between December and June.

The amplitudes of the seasonal cycles of the different constituents vary. Isoprene, HCHO and CO have relatively low intra-annual variation, as their column amounts remain above 50% of their maxima throughout the year. In comparison, methanol, AOD and NO$_2$ exhibit a more extreme seasonality, with values dropping below 30% of their annual maxima. This pronounced dry season peak in the atmospheric constituents is consistent with the burned area seasonal cycle, which rises from $0.02 \pm 0.002\%$ of the study region in July to $0.06 \pm 0.009\%$ in September, before decreasing to $0.02 \pm 0.002\%$ again in October. Burned area values remain below 10% of the September peak for the rest of the year.

The LAI seasonality is substantially different, as values remain above 70% of the maximum throughout the whole year. LAI is elevated at >3 (>90% of the maximum value) between April and October. The regional monthly mean drops slightly (<2.7 or <80%) from December to February. The lower percentage amplitude of the LAI seasonal cycle may be driven by different phenologies of the vegetation in the domain, e.g. Cerrado savanna grassland flowering is relatively consistent throughout the year, while Amazon forest vegetation tends to flower in the dry season (Morellato et al., 2013). Other research in Brazil has found that while seasonal changes in forests are associated with solar radiation, they are driven by rainfall for savannas and grasslands, resulting in opposite cycles for the different land cover types (Myneni et al., 2007). A comparison of 6 LAI datasets consistently shows that in the southern Amazon the LAI in regions of broadleaf forest cover is higher in July, compared to

January, while the savanna/grassland region has the opposite LAI signal (Fang et al., 2013). Consequently, phenology-driven LAI values may peak at different times depending on the land cover types, resulting in full/partial vegetation coverage of the region throughout the year.

The magnitudes of all atmospheric constituents examined in this study increase in the dry season when burned area is largest, suggestive of changes to the atmospheric chemistry due to a substantial pyrogenic source. In particular, the seasonal variability of $NO_2$ and AOD closely matches the seasonality in burned area, while methanol does so with a lag of one month. As outlined above, vegetation cover, as represented by the LAI, shows more consistent values throughout the year. Therefore, constituents with lower seasonal amplitudes are potentially more strongly linked to the seasonality of biogenic emission sources, although other factors such as atmospheric lifetime will have an important impact on the respective atmospheric constituent concentrations.

### 3.3 Spatial distribution of vegetation, fire and atmospheric composition

Across the southern Amazon, broadleaf forest, savanna and grasslands represent the dominant vegetation types. Annual average broadleaf forest cover dominates in the north west (NW), with land surface coverage typically between 80 and 100%, while savannas and grasslands represent the main land cover (80-100% coverage) in the south east (SE) (Fig. 4). In the centre of the domain, a transition region exists where the competing phenologies typically have 40-60% coverage. The corresponding LAI spatial distribution highlights larger values (3.5 to >5.0) in the NW and lower values (2.0-3.5) in the SE. This is consistent with broadleaf forest having a larger biomass (per unit area) in comparison to savanna/grassland vegetation types.

For the dry season burned area data, there are sporadic hot spots peaking at >0.3%. The most coherent spatial structures are a filament of burned area values between 0.2 to 0.3% stretching along the Bolivian eastern border down into Paraguay in the S of the region, and the cluster of western Brazilian burned area values at >0.1%, though peaking at 0.3-0.5%. This latter feature is generally consistent with the reported "arc of deforestation" in the Amazon (Reddington et al., 2015; Santos et al., 2021). The hatching in Figure 4d represents at least a 2.5% decrease in broadleaf forest between 2001 and 2019, which coincides with the burned area patterns reported here. Therefore, the fire activity may be related to regions undergoing deforestation and, to a certain extent, the land cover classifications. For instance, the burned area filament in Figure 4d closely follows the high (>90%)/low (<20%) savanna/grassland and broadleaf forest structure in Figure 4a & b. Thus, it is suggestive of transitional regions between biomes driven by predominately anthropogenic pyrogenic activity. Overall, the spatial distribution of burned area exhibits more localised maxima than vegetation cover.

The spatial distributions of multi-annual mean values of total column isoprene are similar to those of broadleaf forest cover and LAI for both the wet and dry season (c.f. Fig. 5a,b and Fig. 4b,c). Values decrease from the far NW (maximum of 2.7 $\times$ $10^{16}$ molecules cm$^{-2}$ in the dry season) to <0.5 $\times$ $10^{16}$ molecules cm$^{-2}$ in the south (S), following the NW-SE transition from forest to savanna/grassland as evident in the dry season total column isoprene values (c.f. Fig. 4b and Fig. 5b). Over the broadleaf forest region, isoprene is elevated in the west (W), where the maximum forest cover occurs, compared to the east (E). The spatial similarities between broadleaf forest cover and total column isoprene amounts suggest the broadleaf forest is the dominant source of this BVOC for this region, especially in the dry season. This finding is also consistent with the short

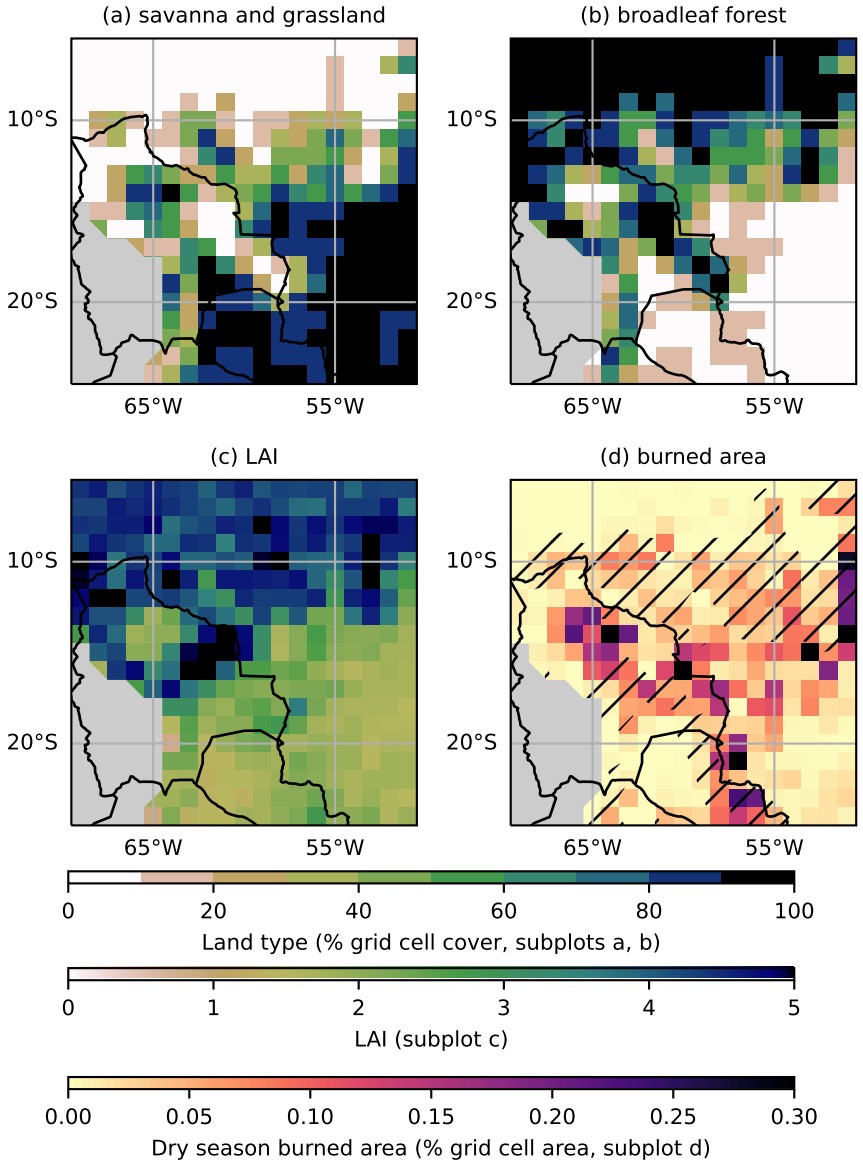

**Figure 4.** The annual mean distribution for 2001-2019 of (a) savanna and grassland and (b) broadleaf forest cover, represented as the % of each grid cell covered by the respective land cover type, and (c) annual mean LAI for 2001-2019. Dry season (August-October) monthly mean burned area (% grid cell area) for 2005-2016 is shown on panel (d) along with regions where at least 2.5% of the area has been deforested marked by hatching. Areas over 1000 m a.s.l. based on the GMTED2010 digital elevation model (Danielson and Gesch, 2011) have been masked on all panels.

lifetime of isoprene confining the peak concentrations to the source region. The W-E gradient in mean total column isoprene over the forested region itself suggests variability in isoprene emissions within the broadleaf forest biome due to different plant species or local environmental conditions, as found by Gu et al. (2017), who identify an elevation gradient in isoprene emissions in the tropical forest N of the region studied here, probably driven by plant species distributions. In this study, the highest total column isoprene values occur at elevations of around 200 m a.s.l., with lower isoprene column densities in the N

of the forest, where elevations decrease to below 100 m a.s.l.

Two other atmospheric constituents have similar distributions in the wet season to that of broadleaf forest cover: CO and AOD. CO is elevated in the NW (>16 $\times$ 10$^{17}$ molecules cm$^{-2}$, maximum of 18.2 $\times$ 10$^{17}$ molecules cm$^{-2}$), compared to the SE (<14 $\times$ 10$^{17}$ molecules cm$^{-2}$). AOD is similarly elevated in the N (>0.2, maximum of 0.35) and reaches a minimum in the SE of the study region (<0.1) (Fig. 6a,c). The spatial patterns of CO and AOD during the wet season are consistent with the

broadleaf forest acting as a source of biogenic precursors of CO and aerosols, as values increase over more densely forested areas (compare with Fig. 4b). This is consistent with biogenic sources having a greater relative impact on CO and aerosols in the wet season, when pyrogenic emissions are minimal (Artaxo et al., 2022).

In contrast, wet season HCHO and NO$_2$ increase from the NW (HCHO and NO$_2$ column values of $\sim$0.8 $\times$ 10$^{16}$ molecules cm$^{-2}$ and <0.6 $\times$ 10$^{15}$ molecules cm$^{-2}$, respectively) to the SE, where HCHO reaches a maximum of 3.2 $\times$ 10$^{16}$ molecules

cm$^{-2}$ and NO$_2$: 1.36 $\times$ 10$^{15}$ molecules cm$^{-2}$. This spatial pattern is not consistent with biogenic emissions from the forest being the dominant controlling factor (Figures 5e and 6e, compare to Fig. 4b). Further, biomass burning is minimal in the wet season (see section 3.2), suggesting another source or sink is driving total column HCHO and tropospheric column NO$_2$ in this season. The wet season methanol concentrations are also different to that of vegetation cover. The highest methanol concentrations of $\geq$0.4 ppbv are recorded in the far E of the domain between 10° and 16° S (Fig. 5c).

The dry season spatial patterns of HCHO, CO and AOD differ somewhat from their wet season distributions, suggesting a change in dominant sources/sinks e.g. from vegetation to fires, transport and/or atmospheric chemistry processes. CO appears more well-mixed compared to the other trace gases as expected with its relatively longer lifetime compared to the other species (average CO tropospheric lifetime of 1-3 months (Seinfeld and Pandis, 2016), compare with Table 1) (Fig. 6b). HCHO, CO and AOD reach maximum values (up to 3.7 $\times$ 10$^{16}$ molecules cm$^{-2}$, 32.7 $\times$ 10$^{17}$ molecules cm$^{-2}$ and 0.88 respectively) over

the transition zone between forests and other land cover classes, although the exact locations of these peaks vary. This region is most strongly affected by deforestation and proximate to the highest average dry season burned areas (compare with Fig. 4). Total column HCHO decreases beyond this zone, consistent with a significant pyrogenic source (Freitas and Fornaro, 2022; Palmer et al., 2007), both further into the broadleaf forest (1.2 to 1.6 $\times$ 10$^{16}$ molecules cm$^{-2}$) and to the S, where values drop to <0.6 $\times$ 10$^{16}$ molecules cm$^{-2}$ (Fig. 5f). Minima in the AOD and CO data are also found in the S in the foot of the Andes and

in the SE (<0.2 and <18 $\times$ 10$^{17}$ molecules cm$^{-2}$, respectively) (Fig. 6d). Peak total column CO is found slightly further S to the HCHO maximum (10-12°S compared to 8-11°S). AOD is characterised by two maxima (>0.8) within the same region as CO, which occur in areas of mixed land cover (forest cover 10-100%) that have experienced substantial deforestation (>2.5% area deforested) and some burning (dry season monthly mean burned area $\sim$0.01%) (compare Figures 6d and 4). However, this does not correspond to the locations of maximum dry season burned areas. The co-location of these maximum values with regions

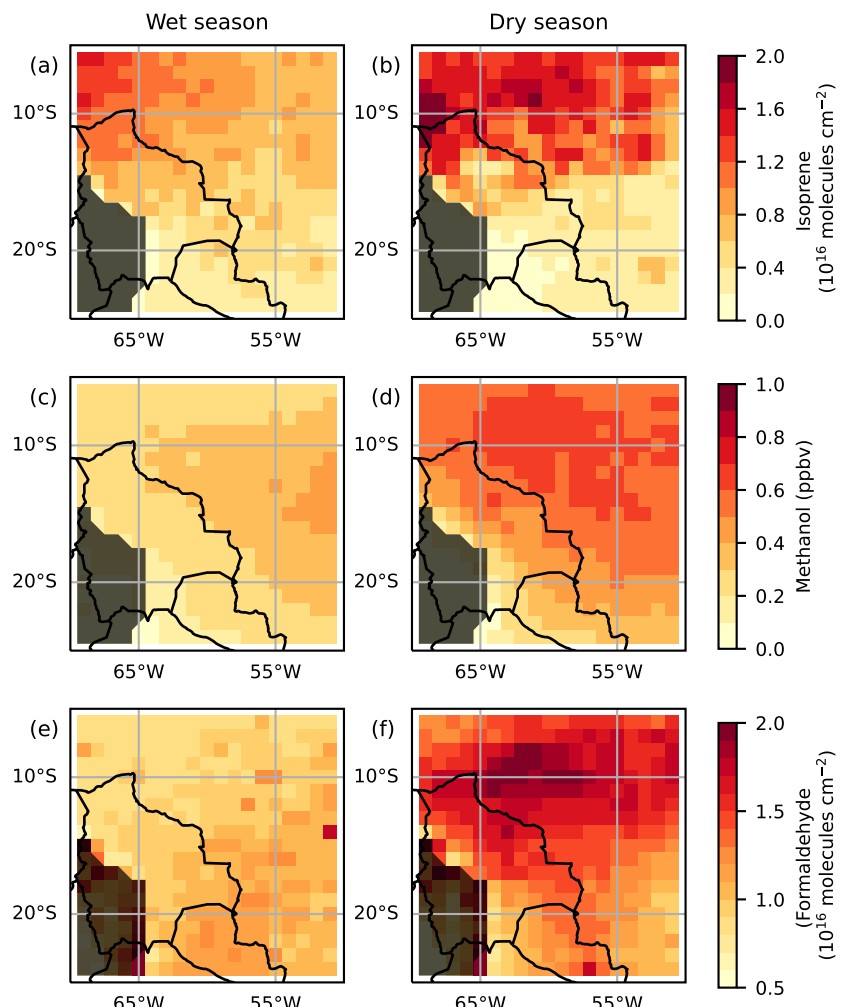

**Figure 5.** Regional distributions of column mean isoprene (top row (a, b), molecules cm$^{-2}$), methanol (middle row (c, d), ppbv), and formaldehyde (bottom row (e, f), molecules cm$^{-2}$, note colorbar starts at $0.5 \times 10^{16}$ molecules cm$^{-2}$) for the wet season (left column) and dry season (right column). The wet season includes the months of February-April, while the dry season covers August-October. The time period varies with constituent. Areas over 1000 m a.s.l. based on the GMTED2010 digital elevation model (Danielson and Gesch, 2011) have been masked on all panels and are not included in further analysis.

of deforestation and biomass burning suggests a pyrogenic source, potentially emissions from deforestation fires, is important for these constituents. These constituents were also elevated to the N of this region where the forests are dominant, suggesting the presence of a biogenic source for this region or transport of pyrogenic emissions. Particularly in the case of HCHO, the

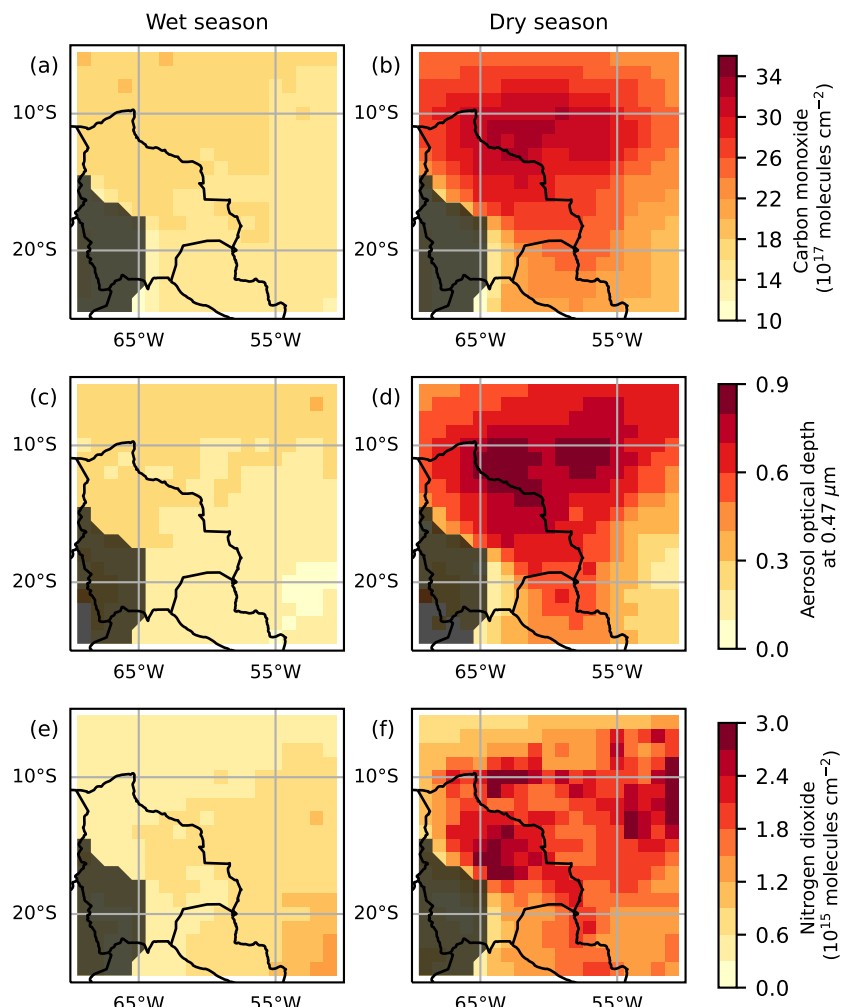

**Figure 6.** Regional distributions of CO (top row (a, b), molecules cm$^{-2}$, note colorbar starts at $10 \times 10^{17}$ molecules cm$^{-2}$), AOD (middle row (c, d)), and NO$_2$ (bottom row (e, f), molecules cm$^{-2}$) for the wet season (left column) and dry season (right column). The wet season includes the months of February-April, while the dry season covers August-October. The time period varies with constituent. Areas over 1000 m a.s.l. based on the GMTED2010 digital elevation model (Danielson and Gesch, 2011) have been masked on all panels and are not included in further analysis.

presence of NO$_x$ associated with biomass burning could affect the HCHO yield from BVOC emissions (Langford et al., 2022), potentially increasing HCHO formation from isoprene oxidation over the broadleaf forest.

Methanol dry season concentration values are similarly elevated (>0.6 ppbv) in the N over the forest to savanna/grassland transition, but they are also consistently high further into the Amazon forest, as well as in the NE. Some of the maxima are

located over areas of mixed vegetation, which experienced substantial deforestation over 2001-2019, as found for HCHO, CO and AOD (compare Fig. 5d with Figures 1 and 4), but other regions do not reflect any of the studied land surface variables.

The dry season $NO_2$ column densities had several distinct maxima $>2.7 \times 10^{15}$ molecules cm$^{-2}$, the highest being $3.5 \times 10^{15}$ molecules cm$^{-2}$, all located 9-16° S. Some of these overlap with the two AOD maxima. Unlike AOD, $NO_2$ is clearly elevated over the maximum burned areas recorded at 15°S, 65°W and in the E of the study region (Fig. 6d,f, Fig. 4d). In regions of minimal fire activity, such as deeper into the Amazon forest and the edge of the Andes, tropospheric column $NO_2$ is $<0.9 \times 10^{15}$ molecules cm$^{-2}$. The spatial pattern of tropospheric column $NO_2$ closely resembles that of burned area, highlighting the close relationship between the trace gas and fire activity, owing to its much shorter lifetime (around 1 day for $NO_x$ at the surface; Jacob (1999)) compared to CO and aerosols.

## 3.4 Influence of vegetation and fire on atmospheric composition

### 3.4.1 Variations in atmospheric composition with broadleaf forest cover

In this section, the variation in atmospheric composition as a function of broadleaf forest cover (i.e. percentage cover in 10% bins) is considered for the dry and wet seasons over the whole region and the time period of co-existing data. Within the dry season, the impact of pyrogenic activity is assessed by splitting the atmospheric constituent data into "low fire" ($\leq 0.04\%$ burned area) and "high fire" ($>0.04\%$ burned area) regimes. Our results are relatively insensitive to the burned area threshold, as well as the bin width choice (see Supplement).

Isoprene consistently increases with higher broadleaf forest cover in all four regimes (Fig. 7a). The spatial distributions rather than temporal variations drive this signal. Values are lower in the wet season (approximately $0.5-0.7 \times 10^{15}$ molecules cm$^{-2}$) than the dry season for broadleaf forest cover values $>20\%$. In densely forested areas (broadleaf forest cover $>90\%$) the dry season isoprene column density is 50% greater than in the wet season at approximately $1.0-1.5 \times 10^{15}$ molecules cm$^{-2}$. The greater change between seasons in isoprene column amounts at higher forest cover suggests that isoprene emissions from broadleaf forest have a stronger seasonality than those from other vegetation types. The differences in mean isoprene column densities between areas of high and low fire activity remain within $0.3 \times 10^{15}$ molecules cm$^{-2}$. Consequently, isoprene responds to the change in forest cover more than the change in burned area, highlighting the importance of its biogenic source, as suggested in section 3.3.

For $NO_2$, a similar pattern occurs with larger column values in the dry season (approximately $1.5-2.0 \times 10^{15}$ molecules cm$^{-2}$) than in the wet season ($\leq 0.86 \times 10^{15}$ molecules cm$^{-2}$) (Fig. 7f). However, the relationship with forest cover is non-linear with peak dry season column $NO_2$ in the 40-50% forest cover bin. When split into the two fire regimes, there is a large column $NO_2$ difference independent of forest cover bin. When fires are low, tropospheric column $NO_2$ ranges between 1.2 and $1.3 \times 10^{15}$ molecules cm$^{-2}$, while in the high fire regime all column $NO_2$ values are $>2.5 \times 10^{15}$ molecules cm$^{-2}$ (i.e. at least 60% larger). Overall, as noted in section 3.3, this result is suggestive of the largest $NO_2$ emissions from pyrogenic sources in forested regions and/or transition zones between vegetation types.

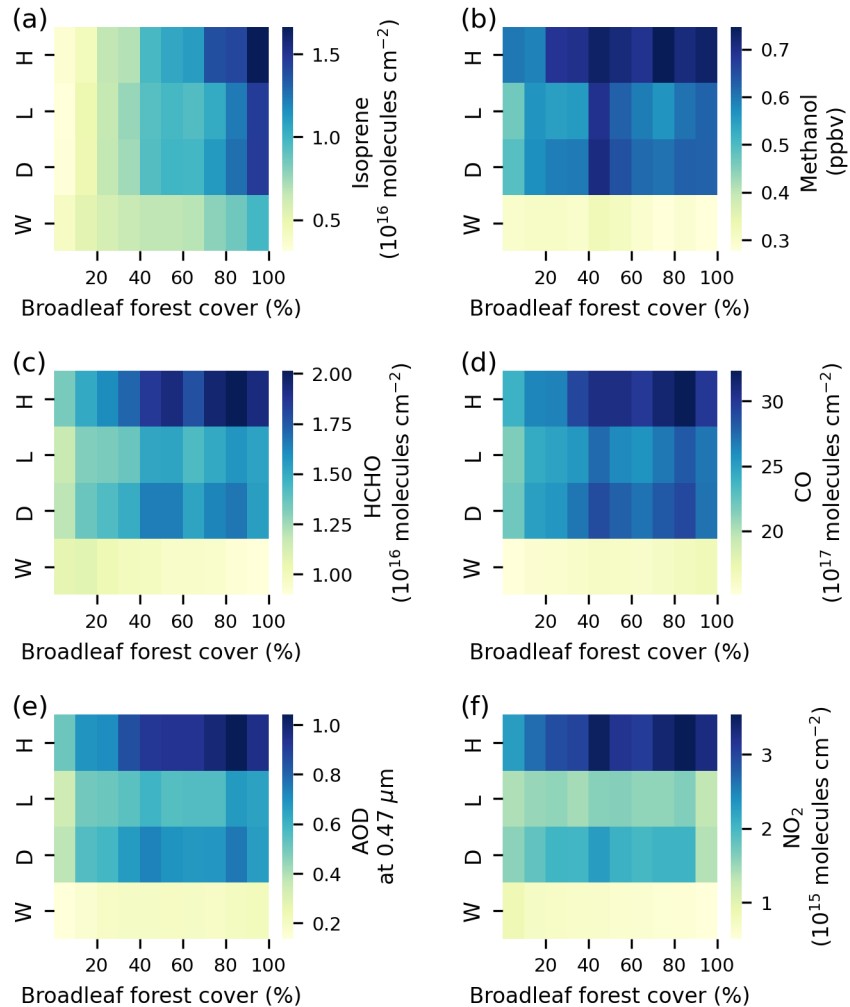

**Figure 7.** Mean monthly atmospheric constituent values depending on percentage broadleaf forest cover in a given grid cell for 4 categories: High burned area dry season (H), Low burned area dry season (L), dry season (D) and wet season (W). Each subplot shows data for one atmospheric constituent: (a) isoprene ($1 \times 10^{16}$ molecules cm$^{-2}$), (b) methanol (ppbv), (c) HCHO ($1 \times 10^{16}$ molecules cm$^{-2}$), (d) CO ($1 \times 10^{17}$ molecules cm$^{-2}$), (e) AOD, and (f) NO$_2$ ($1 \times 10^{15}$ molecules cm$^{-2}$). Each subplot includes data for years when the given atmospheric constituent, land cover and burned area datasets overlap.

Consistent with the other constituents, column methanol values are larger (0.5-0.7 ppbv) in the dry season than wet (<0.3 ppbv). While the low fire regime is very similar to the dry season average, the high fire regime column methanol values are larger ($\geq$0.6 ppbv) for all forest cover bins. Similarly to NO$_2$, though, the peak values are in the 40-50% and 70-80% forest cover bins (Fig. 6b). Therefore, unlike for isoprene, the methanol columns are not linearly linked to forest cover, but typically a larger forest cover percentage bin will have larger methanol values.

The other atmospheric constituents (Fig. 7c,d,e) also have some similarities with both isoprene and $NO_2$. The wet season values are much lower than those in the dry season regardless of forest cover, consistent with the seasonal cycle of the regional averages discussed in section 3.2. In the dry season, HCHO, CO and AOD increase as forest cover increases from 0 to 50% and values remain elevated at higher forest covers ($\geq 1.5 \times 10^{16}$ molecules cm$^{-2}$, $>25.7 \times 10^{17}$ molecules cm$^{-2}$ and $>0.55$, respectively, for forest cover $>50\%$). The high fire regime is associated with higher values of the atmospheric constituents compared to the low fire regime across all forest cover values. HCHO is elevated by at least 13%, AOD by 27%, and CO by 7-17%. The overall maxima are associated with both high burned area and high forest cover of 80-90%. In the dry season average a secondary maximum occurs for the 40-50% (40-60% for HCHO) forest cover bin, resembling that of the peak in $NO_2$ and methanol values. This result suggests both vegetation and fire are important in driving the concentration of HCHO, CO and aerosols in the region in the dry season through a combination of biogenic emissions, particularly from the forest, and pyrogenic emissions in the transition zone and in forested regions, strengthening conclusions drawn from the analysis of spatial maps in section 3.3.

### 3.4.2   Variations in atmospheric composition with burned area

In this section, the change in the atmospheric constituents with burned area is analysed, depending on season (W - wet season, D - dry season) and, for the dry season, dominant land cover (F - $\geq 50\%$ forest cover, S - $<50\%$ forest cover, i.e. $\geq 50\%$ savanna/grassland).

Although isoprene total column amounts are increased during the dry season, when fire activity is at its highest in the southern Amazon (see section 3.2), there is no clear relationship between the burned area extent and the isoprene column amount (Fig. 8a), as the highest dry season total column isoprene values occur at 0-0.01% ($0.9 \pm 0.01 \times 10^{15}$ molecules cm$^{-2}$) and 0.08-0.09% ($0.9 \pm 0.1 \times 10^{15}$ molecules cm$^{-2}$) burned area. The data does further highlight the relevance of land cover type for this trace gas. Regardless of the amount of burning, for each respective burned area bin, total column isoprene is at least 140% higher in regions of dominant forest cover than in regions dominated by savannas and grasslands. Therefore, forest cover is closely connected to isoprene emissions, supporting conclusions reached in section 3.4.1, suggesting the broadleaf forest is the dominant source of this BVOC in the region in the dry season, while the pyrogenic source, limited in its spatial extent, has a more minimal influence, despite the dry season maximum (see section 3.2).

The other constituents are also elevated over the forested area, compared to the savanna/grassland region, although the relative difference in atmospheric constituent between the savanna/grassland and forest categories varies. This difference is particularly pronounced for HCHO and AOD (Fig. 8c,e). HCHO and AOD total column amounts over forests ($1.5\text{-}2 \times 10^{16}$ molecules cm$^{-2}$ and 0.6-1.1) are, respectively, $\geq 24\%$ and $\geq 35\%$ higher than those over savannas/grasslands. $NO_2$ has the smallest difference in values between the land cover categories ($1.4\text{-}3.1 \times 10^{15}$ molecules cm$^{-2}$ for savanna/grassland, $1.3\text{-}4.2 \times 10^{15}$ molecules cm$^{-2}$ for forest) (Fig. 8f). For $NO_2$, AOD, CO and HCHO the differences between land cover types increase at higher burned area values.

The increase in atmospheric constituents over the forest compared to other land cover types may be driven by the emission of biogenic precursors and/or higher pyrogenic emissions when forest, as opposed to savanna/grassland, vegetation is burned. The

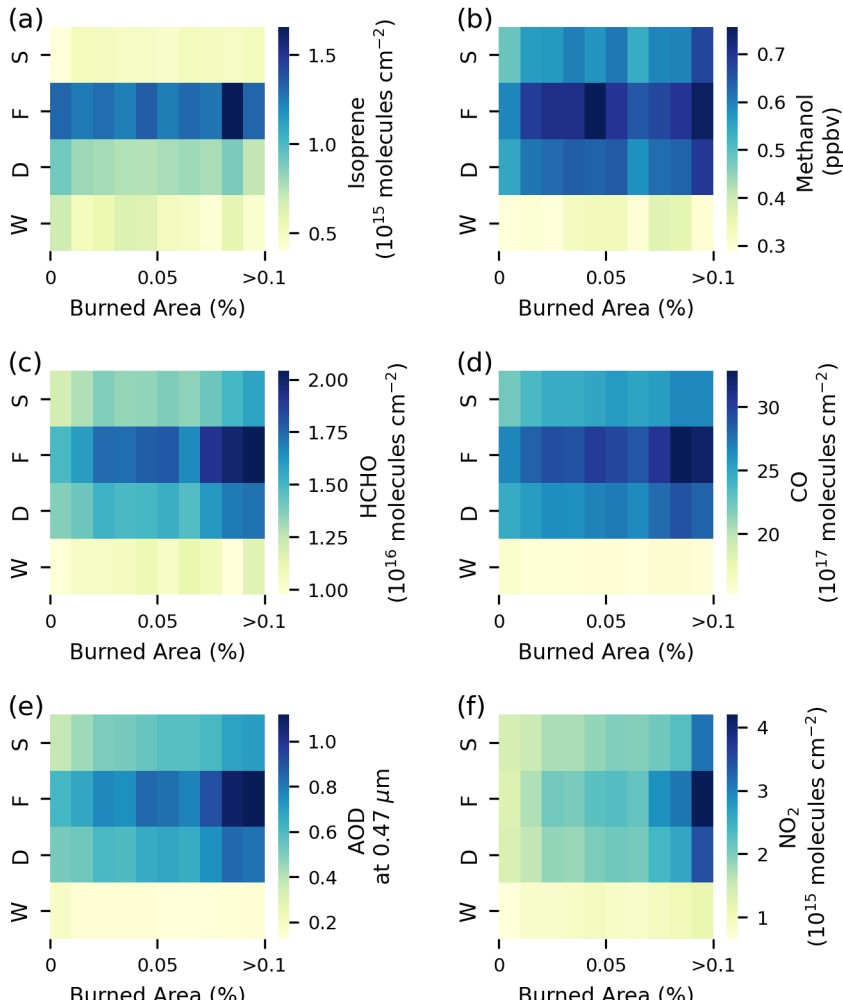

**Figure 8.** Mean monthly atmospheric composition depending on percentage burned area in a given grid cell for 4 categories: Savanna/grassland dominated dry season (S), Forest dominated dry season (L), Dry season (D) and Wet season (W). Each subplot shows data for one atmospheric constituent: (a) isoprene ($1 \times 10^{16}$ molecules cm$^{-2}$), (b) methanol (ppbv), (c) HCHO ($1 \times 10^{16}$ molecules cm$^{-2}$), (d) CO ($1 \times 10^{17}$ molecules cm$^{-2}$), (e) AOD, and (f) NO$_2$ ($1 \times 10^{15}$ molecules cm$^{-2}$). Each subplot includes data for years when the given atmospheric constituent, land cover and burned area datasets overlap.

latter is particularly relevant for NO$_2$, as the difference in column densities between the land cover categories is only significant at high burned area values. At >0.09% burned area, NO$_2$ reaches $4.2 \pm 0.1 \times 10^{15}$ molecules cm$^{-2}$ over forest compared to $3.1 \pm 0.1 \times 10^{15}$ molecules cm$^{-2}$ over savanna/grassland.

On average, most of the same atmospheric constituents (NO$_2$, AOD, CO and HCHO) increase with burned area in the dry season, reaching peak values at 0.08-0.09% burned area (AOD and CO) or >0.09% burned area (NO$_2$ and HCHO). For

**Table 2.** Spearman rank correlation coefficients r (with the Spearman rank correlation coefficient squared: $r^2$, for comparison with Table 3 in brackets). For each correlation, data for the three dry season months for years when the atmospheric constituent and surface variable datasets overlap is used. In the case of annual vegetation data, all months of a given year are assumed to have the same vegetation cover. All r values are significant at the $p = 0.05$ level.

|  | Isoprene | Methanol | HCHO | CO | AOD | $NO_2$ |
|---|---|---|---|---|---|---|
| Broadleaf forest cover | 0.77 (0.59) | 0.19 (0.04) | 0.44 (0.19) | 0.41 (0.17) | 0.39 (0.15) | -0.14 (0.02) |
| LAI | 0.77 (0.59) | 0.22 (0.05) | 0.49 (0.24) | 0.43 (0.18) | 0.38 (0.14) | -0.03 (0.00) |
| Burned area | -0.09 (0.01) | 0.15 (0.02) | 0.23 (0.05) | 0.17 (0.03) | 0.18 (0.03) | 0.46 (0.21) |

methanol the increase with burned area is limited to values of burned area under 0.05%. The greater column amounts of these constituents associated with both forest cover and burned area are consistent with the dry season maxima observed in (or proximate to) forested regions and over burned areas on the spatial maps in section 3.3.

Consequently, all studied atmospheric constituents are influenced by land cover type, as their abundances increase over more densely forested areas. Additionally, maximum values are reached in the presence of burning, especially at $\geq 0.07\%$ burned area for HCHO, CO, AOD and CO in forested regions, showcasing the pyrogenic source. At low burned area values there is little difference between forests and savannas/grasslands for $NO_2$, highlighting that burning is the major source of the trace gas in the region during the dry season and the role of land cover is to modify the emissions where burning occurs.

### 3.5 Statistical relationships between atmospheric composition and land cover, LAI and fire

In this section, statistical relationships between land cover, LAI, and burned area, and the atmospheric constituents for the dry season are explored, to quantify the extent to which land cover variables drive atmospheric composition over the southern Amazon for the time period both datasets are available. Spearman rank correlation (Table 2) and OLS regression (Table 3; see section 2) are utilised. The results from both methods are very similar, with the squared Spearman rank correlation coefficients slightly higher than the equivalent OLS $R^2$ values.

There is a strong significant positive relationship between the two vegetation variables of broadleaf forest cover and LAI, and total column isoprene over the region (Spearman's r = 0.77 for both, OLS $R^2$ = 0.59 and 0.54 for broadleaf forest cover and LAI, respectively), as expected based on sections 3.3-3.4.2. However, there is a weak negative relationship between isoprene and burned area. In contrast, tropospheric column densities of $NO_2$ exhibit a moderate positive relationship with burned area (Spearman's r = 0.46, OLS $R^2$ = 0.25), but weak relationships with land cover variables over the region.

The relationships for the other atmospheric constituents are mixed and much weaker. HCHO, CO and AOD all show positive weak to moderate relationships with both broadleaf forest cover and LAI (r-values around 0.4; OLS $R^2$ = 0.08 to 0.18), suggesting some influence of vegetation on these atmospheric constituents, particularly for HCHO. The relationships between these atmospheric constituents and burned area are considerably weaker, but also positive (r values from 0.17 for CO to 0.23 for HCHO; OLS $R^2$ between 0.01 and 0.03). Methanol showed a similarly weak positive relationship with both land cover and fire (r values from 0.15 with burned area to 0.22 with LAI; OLS $R^2$ between 0.01 and 0.04). These weak results, as compared

**Table 3.** OLS regression coefficients of determination ($R^2$) for the dry season for single and multiple linear regression calculations. For each regression, data for the three dry season months for years when the atmospheric constituent and surface variable datasets overlap is used. In the case of annual vegetation data, all months of a given year are assumed to have the same vegetation cover. All $R^2$ values are significant at the p = 0.05 level. The equivalent scatter plots for the data used to derive the linear OLS coefficients for broadleaf forest cover vs. isoprene and burned area vs. $NO_2$ are shown in figures 9a and 10, respectively.

|  | Isoprene | Methanol | HCHO | CO | AOD | $NO_2$ |
|---|---|---|---|---|---|---|
| Broadleaf forest cover | 0.59 | 0.02 | 0.14 | 0.12 | 0.08 | 0.00 |
| LAI | 0.54 | 0.04 | 0.18 | 0.14 | 0.09 | 0.00 |
| Burned area | 0.02 | 0.01 | 0.03 | 0.01 | 0.02 | 0.25 |
| Forest + Burned Area | 0.59 | 0.04 | 0.19 | 0.14 | 0.11 | 0.25 |
| LAI + Burned Area | 0.54 | 0.05 | 0.23 | 0.15 | 0.12 | 0.26 |

to the isoprene relationship with vegetation, highlight that with the longer lifetimes of these trace gases and aerosols, especially CO and methanol (average lifetimes of 1-3 months and 5 days, respectively, Seinfeld and Pandis (2016); Bates et al. (2021)), there is more transport and mixing.

Multiple OLS regression was also performed to test whether the combination of biogenic and pyrogenic sources improved their explanatory power for the variation in atmospheric constituents. The $R^2$ value is unchanged for isoprene, while for $NO_2$ the $R^2$ values are minimally affected, suggesting one main emission source of vegetation and fire, respectively, for these two atmospheric species. For the other atmospheric constituents, the combination of broadleaf forest cover or LAI and burned area increased the $R^2$ values (as compared to the OLS linear regression results) modestly. The change is most notable for HCHO ($R^2$ value of 0.23 compared to 0.18) and then for AOD ($R^2$ value of 0.12 compared to 0.09). While the overall $R^2$ values are still much lower than the $R^2$ value for the linear relationship between isoprene and broadleaf forest cover, for HCHO the multiple regression $R^2$ values are only slightly lower than for the $NO_2$-burned area relationship. This illustrates the relevance of both a biogenic and pyrogenic source to HCHO column densities, as previously observed elsewhere in Brazil (Freitas and Fornaro, 2022).

The more robust relationships between broadleaf forest cover vs. isoprene and burned area vs. $NO_2$ were studied in more detail (Fig. 9a, Fig. 10). The dry season composition data were binned based on broadleaf forest cover (for isoprene) or burned area (for $NO_2$). We used a bootstrapping approach to test the significance of the isoprene ($NO_2$) increase with broadleaf forest cover (burned area) and found the results are significant at the 95% confidence level (not shown). Over the region, both binned (at 10% broadleaf forest cover intervals) and non binned data suggest that isoprene increases by $1.1 \times 10^{15}$ molecules cm$^{-2}$ for every 10% increase in broadleaf forest cover in the dry season (OLS $R^2$ = 0.59, WLS $R^2$ = 0.97) (Fig. 9). Consequently, the values for total isoprene column densities over completely (100%) forested regions are on average 4 times greater than over non-forested regions (0%). In non-forested regions, isoprene concentrations reflect the local background arising from emissions from non-forest species as well as mixing and transport of forest-related emissions on short time-

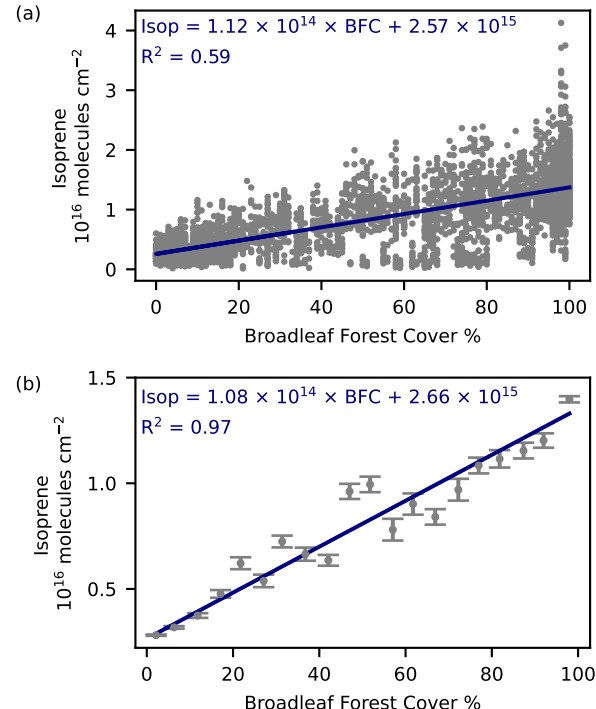

**Figure 9.** The relationship between dry season monthly isoprene data and broadleaf forest cover (BFC) for 2012-2016 with the best fit ordinary least square (OLS) regression line (a). Dry season isoprene data for the same time period binned based on 10% broadleaf forest cover intervals with weighted least squares (WLS) regression results. The errorbars show the standard error for each land cover bin (b). On panel (a) and (b) the text gives the best fit linear regression equations.

scales. Tree species composition, in addition to forest dynamics and environmental conditions affecting the plant's emission efficiency, could explain the variability of isoprene column amounts within each forest cover bin.

The linear burned area/$NO_2$ relationship was found to exhibit different sensitivities depending on vegetation cover (Fig. 10). Over regions of high forest cover and high LAI values, tropospheric $NO_2$ increases with burned area more than over locations with low forest cover and lower LAI (i.e. those dominated by savannas and grasslands, see section 3.4.2). These different

sensitivities are particularly clear up to around 0.5% burned area (Fig. 10, see Fig. 8f for average conditions for burned areas of 0 to >0.1%). Extremely high burned areas tend to occur in less forested regions, though the spatial distributions (Fig. 4) suggest these are savanna/grassland fires proximate to the broadleaf forest in the central part of the region. The dominance of savanna/grassland at extremely high burned areas could decrease the variation in $NO_2$ explainable by forest cover over the whole dataset (Tables 2 and 3).

Hence, to represent this more complex burned area/$NO_2$ relationship we also explored the log-log relationship for $NO_2$ ($\log_e(NO_2)$) and burned area ($\log_e(BA)$) ($R^2 = 0.32$ for non binned data; WLS $R^2 = 0.90$ for binned data (at 0.25 $\log_e(BA)$

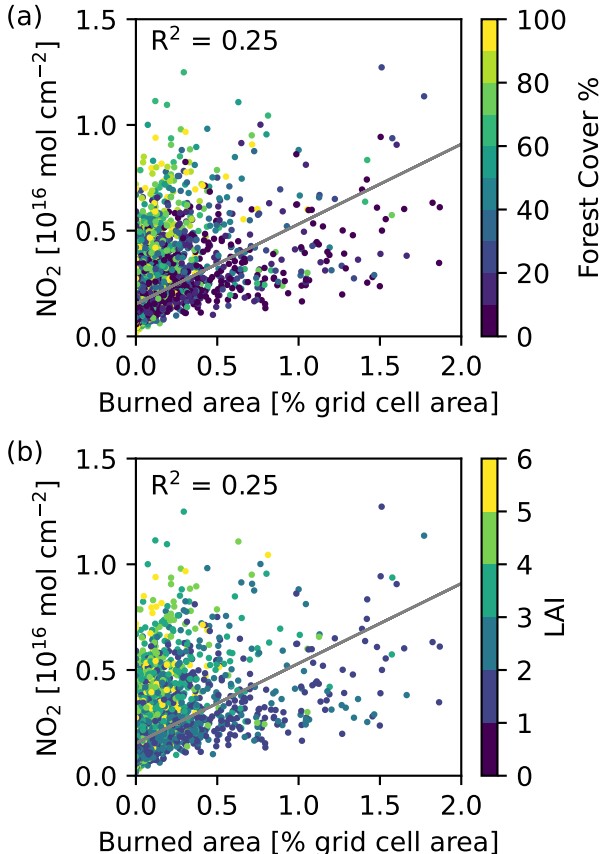

**Figure 10.** The distribution of monthly dry season tropospheric NO₂ columns against monthly dry season burned area % coloured by broadleaf forest cover (a) and LAI (b) for 2005-2019. The panels display the OLS regression line and associated $R^2$ value, as included in Table 3.

intervals); Fig. 11). This log-log relationship, approximately a fifth root power law, indicates that the change in tropospheric column NO₂ is slightly greater at low burned area percentages compared to at high burned areas. This is consistent with the findings for the different vegetation types outlined above. The relationship was found to remain relatively consistent through time by testing the relationship between the natural logarithm of burned area from GFED5 (Chen et al., 2023) and natural logarithm of the NO₂ tropospheric column for the later time period 2017-2020 (not shown). For both NO₂ and isoprene, the WLS linear regression results were largely unaltered when the bin sizes were halved or doubled (not shown).

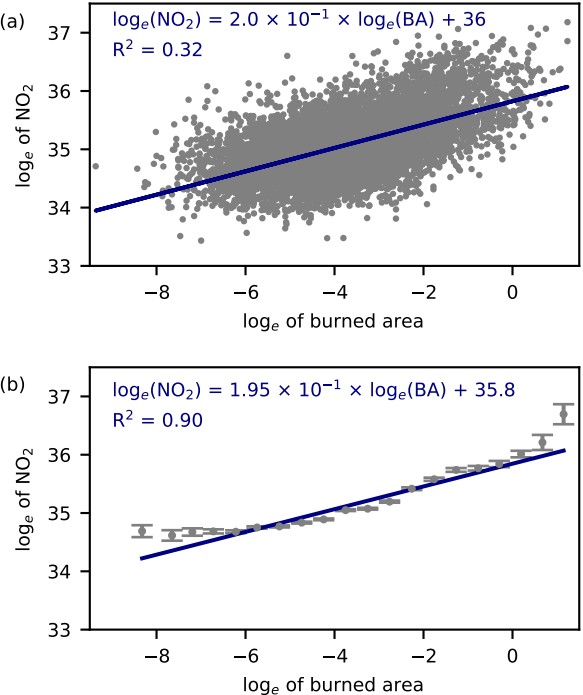

**Figure 11.** Natural logarithm of the monthly dry season $NO_2$ data ($\log_e(NO_2)$) against the natural logarithm of the monthly dry season burned area ($\log_e(BA)$) for 2005-2016 with the results of OLS regression (a). The WLS regression analysis on the natural logarithm of monthly dry season $NO_2$ data binned based on the natural logarithm of monthly burned area percentage cover for the same time period (b). The errorbars show the standard error for each burned area bin. On panel (a) and (b) the text shows the best fit equation found by the regressions and the respective $R^2$ value.

## 4 Discussion and conclusions

This study used new and complementary satellite observations to investigate the relationships between land cover, LAI, and burned area, and five trace gases (isoprene, methanol, HCHO, CO, $NO_2$) and AOD in the southern Amazon, with the aim to understand the influence of vegetation and fire on atmospheric composition in a region of land cover change.

Biomass burning is identified as a potential driver of year-to-year variability in several trace gases: CO, HCHO and $NO_2$, as well as aerosols, as suggested by the AOD record. Previous studies have identified drought years in 2005, 2007, 2010, 2012 and 2015 for this region (Pope et al., 2020; Panisset et al., 2018; Reddington et al., 2015), which may increase burning and modify the relationship between burned area and emitted trace gases and aerosols. Although 2007 and 2010 are clearly elevated for burned area, AOD, CO, $NO_2$ and HCHO, not all of the listed drought years show maxima for a given variable in this study.

Pyrogenic sources in the dry season may further drive the seasonal cycle maxima of most of the atmospheric constituents over the region. The observed mean monthly isoprene seasonal cycle differs from the other constituents with a likely vegetation-

driven secondary peak in the wet season. The isoprene seasonality is consistent with greater photosynthetically active radiation (PAR) in the dry season, as well as potentially greater isoprene emissions in response to water stress (Yáñez-Serrano et al., 2020). The isoprene minimum observed here in June has been previously linked to new leaf growth in the tropical forest in the transition between the wet and dry seasons, as young leaves start producing isoprene only after a few weeks (Barkley et al., 2009). The dry season peak for HCHO in September is also consistent with a previous satellite study; however, a secondary wet season peak was found in that work (Barkley et al., 2009). A weaker relationship between isoprene and HCHO is suggested for this region, since HCHO does not appear to respond to the variations in isoprene at the end of the wet season.

The spatial distributions across the southern Amazon further highlight the co-location of isoprene with broadleaf forest cover and high LAI, and $NO_2$ with burned area. Most of the other atmospheric constituents show characteristics of both distributions. Methanol is unusual, due to a greater region of elevated column values in the N and NE of the region.

Broadleaf forest cover (LAI) explains 59% (54%) of the variation in total column isoprene across the study region for the dry season over 2012-2019, which increases to 97% when the isoprene data is binned based on 10% forest cover bins. In the dry season, isoprene column amounts increase linearly with tree cover. For every 10% increase in broadleaf forest cover the isoprene total column amount increases by $11 \times 10^{15}$ molecules cm$^{-2}$, or around 40% of the average background (0% broadleaf forest cover) total column isoprene. This result is consistent for both the binned and original data.

In contrast, burned area explains 25% of the variability in tropospheric column $NO_2$ for the dry season. 32% of the $NO_2$ variability can be explained when the natural logarithms of each variable are used, and the relationship is well-captured when this data is binned at 0.25 $log_e$(BA) intervals ($R^2$ = 0.90). There is a stronger increase in the trace gas at lower burned area values, which is captured by the fifth root power law. Additionally, the $NO_2$ amount varies depending on burning location, with greater values of $NO_2$ for an equivalent burned area in regions with at least 50% broadleaf forest cover. These results are in agreement with the FINNv2.5 emissions inventory, which has larger biomass burning emission factors for tropical forest compared to savanna/grassland (Wiedinmyer et al., 2023). However, the burned areas can be greater where tree cover is more sparse, highlighting the potential for substantial pyrogenic emissions from both forest and savanna/grassland regimes.

In the wet season low tropospheric column $NO_2$ values over the forested region and higher values in the SE could be driven by the forest canopy acting as an $NO_2$ sink through biological uptake, as suggested by Kang et al. (2023). Alternatively, a further $NO_x$ emission source e.g. from soils associated with agricultural activities (Wong and Geddes, 2021) or long-range transport of anthropogenic emissions in the region of São Paulo (van der A et al., 2008), may influence $NO_2$ concentrations.

The clear dry season relationships of isoprene with broadleaf forest cover and $NO_2$ with burned area contrast with the mixed signals for the other atmospheric constituents. However, the combination of broadleaf forest cover and burned area can explain 23% of the variation in dry season total column HCHO, suggesting interactions between pyrogenic and biogenic emissions of HCHO and its precursors. The moderate correlation values of AOD and CO with the land cover variables suggest some influence of vegetation through a biogenic source that yields CO and SOA formation in August-October, although forest emissions are most relevant in the wet season (Yáñez-Serrano et al., 2020; Artaxo et al., 2022). Methanol does not exhibit a strong relationship with any land surface variable despite some observed similarities.

A key difference between methanol, AOD, CO, and isoprene, HCHO and $NO_2$ is their lifetime. While isoprene, HCHO and $NO_2$ have lifetimes of up to a day (Pacifico et al., 2009; Wells et al., 2020; Pommier, 2023; Jacob, 1999), aerosols,

methanol and CO atmospheric lifetimes range from several days to months (Bates et al., 2021; Hodzic et al., 2016; Holloway et al., 2000). The extended time period the aerosols, methanol and CO are present in the atmosphere will increase the role of transport in the observed distribution, resulting in a less clear local source signal. These longer lived species, particularly CO and aerosols, can therefore be transported from more distant anthropogenic sources (e.g. Park et al. (2015); Wang et al. (2015)). However, anthropogenic emissions are thought to be minor compared to pyrogenic and biogenic sources in the study region.

Anthropogenic sources are in the SE of the study area and beyond the region of interest, e.g. the large agglomerations of São Paulo and Rio de Janeiro located further to the SE (see e.g. the European Commission EDGAR v6.1 Global Air Pollution Emissions database, last access: December 2023).

The findings confirm the tropical broadleaf forest, as opposed to other vegetation types, as the dominant source of isoprene in the region, consistent with tropical trees being the dominant source of isoprene globally (Guenther et al., 2012). $NO_2$ is

predominantly driven by pyrogenic emissions in the dry season, although the land cover type modulates the emission amount. HCHO, and to a lesser extent CO and aerosols, is linked to both biogenic and pyrogenic drivers. More specific land cover categories, and/or a consideration of other factors are necessary to identify the potential biogenic sources of methanol. The study finds both land cover and fire have significant impacts on regional atmospheric composition in the southern Amazon, including modifying amounts of trace gases and aerosols that have implications for regional air quality. Therefore, having

established the importance of vegetation and fire activity on South American atmospheric composition, future work could exploit these relationships for Earth System Model (ESM) evaluation, and by using ESMs to explore the underpinning processes and potential feedbacks between the biosphere and atmosphere.

*Code and data availability.* The code used to calculate statistics on the data and produce the plots is shared in the github repository: "https://github.com/sands-eg/SouthernAmazon_figures". The isoprene data is available on request from Kelley Wells and others at the Univer-

sity of Minnesota. The monthly methanol data used in this study is available on Zenodo (https://doi.org/10.5281/zenodo.11472103). The level 2 HCHO data is available for download from https://disc.gsfc.nasa.gov/datasets/OMHCHO_003/summary. The Mopitt CO data products and the MOD08_M3 product containing the AOD data are available for download through the Earthdata portal (https://www.earthdata.nasa.gov/). OMI $NO_2$ data is available for download from www.temis.nl.

*Author contributions.* RP, RMD, FOC, and ES designed the research study. ES and RP analysed the satellite data supported by advice from

and constructive discussion with RMD, FOC, CW, and HP. ES prepared the manuscript with scientific, editorial and technical input from all coauthors.

*Competing interests.* At least one of the (co-)authors is a member of the editorial board of Atmospheric Chemistry and Physics.

*Acknowledgements.* This research has been supported by the Natural Environment Research Council (NERC) through a SENSE CDT studentship (grant no. NE/T00939X/1) and through funding for the National Centre for Earth Observation (NCEO, award reference NE/R016518/1).

This work used JASMIN, the UK's collaborative data analysis environment (https://jasmin.ac.uk). For the methanol data, RAL's NRT system processes Eumetsat Level-1 data from MetOp-B IASI, MHS, AMSU, and GOME-2 and uses ECMWF meteorological forecast data, all processed on RAL's Jasmin infrastructure. We acknowledge the free use of tropospheric $NO_2$ column data from the OMI sensor from www.temis.nl. The isoprene data was provided courtesy of Kelley Wells and others at the University of Minnesota. We acknowledge the advice and support from Brian Kerridge, Richard Siddans, Lucy Ventress and Barry Latter from the Rutherford Appleton Laboratory when

using the IASI methanol data in this study.

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
