# Peer review of "Satellite-observed relationships between land cover, burned area and atmospheric composition over the southern Amazon."

_EGUsphere, 2024_

## Referee Comment (RC2)

**Review for Sands et al. (2024)**

**Article title:**

**Satellite-observed relationships between land cover, burned area and atmospheric composition over the southern Amazon.**

In this article, Sands et al. investigate the link between land surface changes, particularly in South America's rainforests, and atmospheric species of biogenic and pyrogenic sources. It focuses on the following atmospheric species: Isoprene, methanol, formaldehyde HCHO, carbon monoxide CO, and nitrogen dioxide $NO_2$. The land variables studied are land cover type and Leaf Area Index LAI. Different satellite instruments and other data sources are used in the study. For instance, land cover type, Leaf Area Index LAI, and Aerosol Optical Depth AOD are (from MODIS, Aqua and/or Terra); burned area (from GFED4 inventory), Isoprene (from CrIS, Suomi-NPP), Methanol (from IASI, Metop-B), Formaldehyde HCHO and Nitrogen Dioxide $NO_2$ (from OMI, EOS Aura), and carbon monoxide CO (from Mopitt, Terra). The period covered is within 2000-2020, however, some of the data are only available since 2001, 2005, 2008, and 2012; and some of them are unavailable after 2016, 2018, and 2019.

Overall the study is well conducted. The analysis is conducted on different layers, temporal (seasonal and yearly analyses), spatial, and per land cover type and leaf coverage. Statistical methods were also used to draw clearer conclusions as needed. This study is suitable for the journal. No major changes are needed, only minor improvements that will make it flow better to the reader. The analysis presented in this study is useful for the improvement of Earth System Models.

**Specific comments:**

- Metop-B was launched in 2013, this means that the IASI data you got from 2008 until 2012 are from Metop-A?
- It is better to add a *data availability* section at the end of the article, (after the code availability for instance), in which you list the data you used with the corresponding links or sources where we can download them, if possible.
- In the caption of Fig. 5, try to be consistent with the caption of Fig. 4 when mentioning the shaded area in grey.
- It would help is the statistical methods are explained in more details, with equations for instance.
- Add more results in the abstract, to highlight the main findings of the study. The results mentioned in the conclusions for instance, they can be rewritten in a more concise way in the abstract.

**Minor Comments per line:**

- Line 14-15: put a comma before and after these words "such as deforestation"
- Line 18: Maybe cite some of the "particular development stages"
- Line 21: the sentence seems like it is chopped "uncertainties in its magnitude remain.". Re-write or continue the sentence.

- Line 26: for every atmospheric species you write the chemical formula, you add here for isoprene (C5H8) for instance. Like you do for methanol.
- Line 56: you wrote particula**r** matter instead of particula**te** matter, also you can add ($PM_{10}$ and/or $PM_{2.5}$)
- Line 105: You mention that the data product has been quality assessed (Fu et al., 2019; Wells et al. 2020, 2022). Can you add a sentence to tell us the result of these assessments?
- Line 108: you add here the altitude of the Metop-B satellite, just like you did for OMI, EOS Aura.
- Line 110: add *(EOS)* after *Earth Observing System*.
- Line 178: change -average to average without the "-" in the beginning.
- Line 255: mention the lifetimes of other species when you compare the lifetime of CO to them. You do this later in the article, but it is helpful to have the lifetimes of other species mentioned earlier.
- Line 389: consider moving Fig. 9 and Fig. 10 closer to the text where you refer to them. Especially Fig. 10. Or maybe, you can merge them into one figure so it is easier for the reader to read the text, then look at the plots.

---

## Author Comment (AC1)

Dear editor and reviewers,

We thank the reviewers for their valuable feedback. We include the specific and technical comments below (reviewer comments in bold), followed by our responses (regular black font) and text illustrating the changes in the manuscript (in italics, red font for text that has been removed, blue for added text). We have numbered the reviewer comments to help with the ease and clarity of the response, e.g. in cases where both reviewers had similar feedback. Line numbers in the reviewer comments are from the submitted manuscript, while the line numbers in the responses refer to the tracked changes document.

**Reviewer 1:**

Specific Comments :

**1. You should list all the gases / metrics examined in abstract and methanol results should be mentioned**

Thanks for pointing out this omission. We have updated the abstract to include a more explicit list of all the metrics examined. We further include the methanol results and expand on the AOD and CO results (also relating to comment 42 from reviewer 2).

Revised abstract (lines 1 to 15):

Land surface changes can have substantial impacts on the interactions between the biosphere-atmosphere interactions. In South America, rainforests abundantly emit biogenic volatile organic compounds (BVOCs), which coupled with pyrogenic emissions from deforestation fires, can have substantial impacts on regional air quality. We use novel and long-term satellite records to characterise the impacts of biogenic and pyrogenic emissions on atmospheric composition for the period 2001 to 2019 in the southern Amazon, a region of substantial deforestation. These records include those of five trace gases: isoprene ( $C_5H_8$ ), formaldehyde (HCHO), methanol (CH3OH), carbon monoxide (CO), and nitrogen dioxide (NO2); aerosol optical depth (AOD); vegetation (land cover and leaf area index); and burned area. We find that The seasonal cycle for all of the atmospheric constituents peaks in the dry season (August-October) and year-to-year variability in carbon monoxide CO, formaldehyde- HCHO, nitrogen dioxide NO2, and AOD is strongly linked to burned area. We find a robust relationship between broadleaf forest cover and total column isoprene  $C_5H_8$  ( $R^2=0.59$ ), while burned area exhibits an approximate 5th root power law relationship with tropospheric column NO2 ( $R^2$ =0.32), both in the dry season. Vegetation and burned area together show a relationship with HCHO ( $R^2$ =0.23). Wet season AOD and CO follow the forest cover distribution. The land surface variables

are very weakly correlated with CH3OH, suggesting other factors drive its spatial distribution. Overall, we provide a detailed observational quantification of biospheric process influences on southern Amazon regional atmospheric composition, which in future studies can be used to help constrain the underpinning processes in Earth System Models.

**2. Flow of the intro could be improved (there are several short paragraphs, and some choppy sentences)**

We thank the reviewer for this suggestion. We have merged some of the short paragraphs, included a discussion of fires and deforestation in the region (as suggested in reviewer 1 comment 24) and edited individual sentences to improve the flow of the introduction. The tracked changes manuscript includes the updated text of the introduction.

**3. Consider listing the BVOCs you will be focusing on when you introduce what BVOCs are**

We have updated the sentence, lines 31-34, to highlight that isoprene and methanol are the dominant biogenic trace gases studied in the paper:

In the Amazon rainforest, isoprene and methanol (CH3OH), both analysed in this study, are the most strongly emitted biogenic compounds based on mixing ratio measurements (Yáñez-Serrano et al., 2020).

**4. Define emission factor**

We have added text defining emission factors to line 28:

This has driven the use of emission factors, empirically-derived values used to scale calculated trace gas emissions, to describe the sensitivity of emission strength to plant type (e.g. Guenther et al., (1995, 2012); Pacifico et al., (2011); Weber et al., (2023)).

**5. In intro or methods, the use of AOD should be explicitly defined in the context of its use in the paper and how it relates to fires**

We have added the following text introducing AOD and how it can be affected by pyrogenic emissions in section 2.2.2 (Atmospheric composition data), lines 155-158:

AOD is a measure of light attenuation by atmospheric aerosols due to either absorbance or reflectance (Wei et al., 2020). Low AOD values (<0.1) indicate clear sky and low aerosol amounts, while values of 1 suggest very hazy conditions with high aerosol concentrations. Consequently, an increase in aerosol concentration,

**e.g. due to emissions of particles during combustion, should lead to higher AOD values.**

**6. Given the repeated reference to lifetimes, a table of these would be helpful**

Thank you for this suggestion. We have added a 'lifetime' column to Table 1., which provides a broad classification of the lifetimes of the trace gases and secondary organic aerosols.

**Table 1.** Summary of satellite datasets analysed in the study. All datasets were analysed on  $1^{\circ} \times 1^{\circ}$  spatial resolution. Lifetimes are taken from Jacob (1999); Holloway et al. (2000); Pacifico et al. (2009); Hodzic et al. (2016); Wells et al. (2020); Bates et al. (2021); Pommier (2023)

| Variable                            | Source                | Temporal Period | Period of Data | Lifetime        |
|-------------------------------------|-----------------------|-----------------|----------------|-----------------|
| Land Cover                          | MODIS, Terra and Aqua | Annual          | 2001-2020      | NA              |
| Leaf Area Index (LAI)               | MODIS, Terra          | Monthly/Annual  | 2001-2020      | NA              |
| Burned area                         | GFED4.1s              | Monthly         | 2001-2016      | NA              |
| Isoprene                            | CrIS, Suomi-NPP       | Monthly         | 2012-2020      | <1 day          |
| Methanol                            | IASI, Metop-A/B       | Monthly         | 2008-2018      | Days-Months     |
| Formaldehyde (HCHO)                 | OMI, EOS Aura         | Monthly         | 2005-2018      | <1 day          |
| Carbon Monoxide (CO)                | Mopitt, Terra         | Monthly         | 2001-2019      | Months          |
| Nitrogen Dioxide (NO 2 ) | OMI, EOS Aura         | Monthly         | 2005-2020      | <1 day          |
| Aerosol Optical Depth (AOD)         | MODIS, Terra          | Monthly         | 2000-2019      | SOA: Days-Weeks |

**7. The methods should provide more details on how the comparison between areas is made as much of the analysis relies on comparing characteristics of different domains.**

We have added text explaining the comparison between different domains in more detail to section 2.3 (lines 183-187).

Whenever data for different domains is compared (e.g. difference in isoprene over regions of low and high burned area at a given forest cover), the mean across all available data satisfying the domain conditions was calculated. First, the relevant datasets were subset in time so the only years included were those for which land cover, burned area and composition data are all available. Next, the data were split based on the burned area threshold. Each subset (high and low burned area) was then sorted into land cover bins, then the mean and standard error for each bin were calculated.

**8. Give mention of the associated uncertainties in the various data products.**

Thanks for pointing out this omission. The following statements were added in sections 2.2.1 and 2.2.2 to address the uncertainties in the various data products:

- Lines 106/107: The C6 Modis land cover product is assessed to have an accuracy of 73.6% (for the primary land cover classification (Sulla-Menashe et al., 2019).
- Lines 108-111: The LAI product is based on measurements from the Terra satellite at an 8-day temporal resolution and, validated against ground measurements, has a root mean square error of 0.69 across all biomes (Devadiga and Nickeson, 2023).
- Lines 115-119: For the period of interest the GFED4.1s burned area is predominantly based on the MCD64A1 product, which has been found to have a 68% burned area omission error (Padilla et al., 2015). However, GFED4.1s includes the addition of small fire burned areas, which likely counter some of the omission error, as the GFED4.1s burned area is 37% greater than that of GFED3, which did not include small fire estimates (van der Werf et al., 2017).
- Line 127: the retrieved isoprene column amounts differ by 20% to 50% compared to ground-based column measurements
- Lines 133-136: Pope et al. (2021) found a systematic difference of around 30% compared to the Atmospheric Tomography Mission (ATom) flight measurements in areas of methanol enhancement, as well as an uncertainty of 40% to 50% for an individual sounding, which will have been reduced here by averaging.
- Lines 143-144: Uncertainties of individual retrievals of the HCHO columns range within 50% to 105%, with HCHO hotspots characterised by lower uncertainty values, and averaging leading to uncertainty reduction (OMI Team, 2012).
- Lines 152-154: Boersma et al. (2011) estimated the uncertainty for individual retrievals of the NO2 tropospheric columns to be 1.0 x 1015 molecules cm-2 + 25% of the retrieval.
- Lines 162/163: AOD retrievals are expected to have errors within ±0.05 + 0.2
  × AOD value (Levy et al., 2013, Sayer et al., 2014).
- Lines 167-169: The CO total column values have biases of less than 0.2 x 1018 molecules cm-2 and standard deviations of around 0.2 x 1018 molecules cm-2 compared to NOAA validation sites and a field campaign (Deeter et al., 2017).

**9. The abstract describes using novel data sets –make reference to this in the dataset description**

We have edited the sentences on lines 125 and 129 to highlight the isoprene and methanol data products as being novel data sets:

Isoprene column densities were calculated using two isoprene infrared absorption features in the spectral range 890-910 cm-1 and thethis novel data product has been quality assessed (...).

and

The total column methanol data, another recently developed dataset, comes from the Infrared Atmospheric Sounding Interferometers (IASI) on board the MetOp-A and MetOp-B satellites.

**10. Line 15 : specify if this is an average value.**

We have added 'on average' to the sentence to clarify (line 17).

*Ten million hectares of forest on average were cut down globally each year over 2010-2020.*

11. Line 21 : " uncertainties in its magnitude remain" ... consider rewording to indicate broad estimates are a result of uncertainties in the sources / sinks, etc. ...

We have replaced "*highlighting that substantial uncertainties in its magnitude remain*" with ". *The large range is predominantly driven by uncertainties in the emission rates from different plant functional types (Szopa et al., 2021)*" on lines 24-25.

12. Line 32 : "However, the pyrogenic source of HCHO is more significant than the pyrogenic emission of isoprene, and it is an oxidation product of many other gases," which is an oxidation product?

We have split the sentence in lines 38-40 to clarify the focus on HCHO.

However, the pyrogenic source of HCHO is more significant than the pyrogenic emission of isoprene<del>, and it.</del> Further, HCHO is an oxidation product of many other non-biogenic gases, introducing challenges to the interpretation of the data

**13. Line 38 – "We utilise these measurements" – be specific**

We have modified the sentence in lines 45-47 as follows for specificity.

In this study, we analyse these satellite-derived datasets of column isoprene alongside the more established HCHO product <del>We utilise these measurements</del> <del>alongside HCHO in this study</del> to quantify vegetation-driven changes in composition in the southern Amazon (see section 2.1).

**14. Line 69 : add in years of study focus to introduction summary**

We have added the years of study focus to the introduction summary paragraph (line 85).

We compare this range of atmospheric constituents to vegetation and fire proxies using both new and complementary satellite datasets to build a comprehensive picture of the relative impact of both biogenic and pyrogenic sources on regional atmospheric composition during the period 2001-2019.

**15. Figure 1 : the box of the analysis region is difficult to make out, it should be made more prominent in some way**

We have increased the thickness of the box boundary and changed the linestyle to dashed to help the study region boundary stand out on Figure 1:

**16. Line 104 : in what way has the data quality been assessed**

We have added the statement "through comparison against ground-based isoprene column measurements in the Amazon" to the sentence on line 126.

**17. Line 146 : provide descriptions and / or equations**

We have added the Spearman rank correlation coefficient equation (line 190-192):

$$r = 1 - \frac{6\sum_{i=1}^{n} d_i^2}{n(n^2 - 1)},$$

*"where r is the Spearman rank correlation coefficient, n is the sample size and di represents the difference between the ranks of the ith values from each sample",*

as well as the following brief descriptions of the OLS and Theil-Sen regression methods in lines 193-196.

The OLS regression aims to minimise the sum of the squared differences between the observations and the model (residual sum of squares) when fitting a linear model. The Theil-Sen regression is more robust to outliers than OLS regression, as it fits a slope and intercept based on the spatial median of parameters calculated on subpopulations of the data.

**18. Line 147 : define threshold for "clear relationship"**

We include a threshold of  $R^2 \ge 0.25$  (line 196).

Where a clear relationship (OLS coefficient of determination  $(R^2) \ge 0.25$ ) was identified between an atmospheric constituent and land cover or burned area, the data were binned by the explanatory variable, and a weighted least squares (WLS) regression was used to account for the variance of the data in each bin.

**19. Line 157 : "display opposite behaviors" wording vague, specify behavior as trend over time and / or reference reader to figure 2a/b**

We have updated the sentence (lines 206-208) as follows.

The two land cover categories: broadleaf forest and savanna/grassland, display opposite behaviours trends through time (Fig. 2a,b), reflecting that broadleaf forest cover is being replaced by the savanna/grassland modal land cover type.

**20. Line 161: define variation – if 3.1 is the mean value then you can include the standard deviation**

The mean value is 3.1 and we have added the standard deviation (lines 210/211).

Average annual LAI values in the region fluctuate around a value of 3.1 with a standard deviation of 0.06. with considerable inter-annual variation.

**21. Line 166: is this the overall variability or the timing of the extremes**

We think, in addition to the timing of the extremes, the interannual variability has decreased since 2011. We have updated the sentence with a corrected value for

monthly average burned area and added 'inter-annual' to the variability statement in line 215.

Since 2011, the monthly average burned area has remained below 200 km2 in the peak burning months and the inter-annual variability has decreased.

**22. Line 189: the figure caption says the values are normalized to 1, but the uncertainty is given as a percent, make this clearer**

We have added 'of the study region' after the first burned area value on line 240 to highlight that this is the percentage of the area that is burned, not the fraction of maximum September burned area.

This pronounced dry season peak in the atmospheric constituents is consistent with the burned area seasonal cycle, which rises from  $0.02 \pm 0.002\%$  of the study region in July to  $0.06 \pm 0.009\%$  in September, before decreasing to  $0.02 \pm 0.002\%$  again in October.

23. Line 220: "Therefore, the fire activity is related to regions undergoing deforestation and, to a certain extent, the land cover classifications." Suggested rewording of this statement, it seems too conclusive to say they are "related" based solely on the observations.

We have reworded the statement (line 272) to be less conclusive.

Therefore, the fire activity **is may be** related to regions undergoing deforestation and, to a certain extent, the land cover classifications.

**24. Line 233: you should discuss "anthropogenic pyrogenic activity" in the introduction and deforestation fires (line 266)**

Thanks for pointing out this omission. We have added a paragraph discussing fires and their links to land cover change in the Amazon region to the introduction (lines 48-56).

The Amazon and neighbouring savannas and grasslands experience significant impacts from fire activity, which have been reviewed by Pivello (2011). Fires in the region have both natural and anthropogenic causes. Lightning can ignite the savanna and grassland vegetation; these ecosystems are fire-dependent, meaning many of the species have adapted to recurrent fires. However, unlike the savanna region, the Amazon rainforest is sensitive to burning and the ecosystem can be destroyed through fire activity. In the 21st century, the majority of wildfires in Brazil have anthropogenic causes, as natural vegetation (e.g. the rainforest) is removed for agriculture. In fact, fire is one of the major causes of land cover change globally (Heald and Spracklen, 2015). This suggests that regions of land cover change driven shifts in biogenic emissions will also often experience substantial pyrogenic impacts on the atmospheric composition.

**25. Line 235 : does the instrument have enough sensitivity to capture a difference in the lowest 100 m**

The paper referred to in the manuscript (Gu et al., 2017) suggests that a shift in the plant species distribution due to higher elevation, may lead to more plants that are higher isoprene emitters. We suggest that the higher isoprene column concentrations in regions with elevations of over 200 m a.s.l. may be driven by a similar change in species distribution. Given sufficient vertical transport, the instrument would not have to be sensitive to the lowest 100 m, but the spatial distribution of high isoprene emitting plants may be observed.

**26. line 343: because this is the start of a new paragraph you should specify what you are comparing the "increase" in constituents to**

We have updated the sentence (line 395) to read:

The increase in atmospheric constituents over the forest **compared to other land cover types** may be driven by the emission of biogenic precursors and/or higher pyrogenic emissions when forest, as opposed to savanna/grassland, vegetation is burned.

**27. line 471: statement indicates confirming a finding, include reference(s).**

We have added a reference to Guenther et al., 2012, who identify tropical trees as the major source of isoprene globally (line 523).

The findings confirm the tropical broadleaf forest, as opposed to other vegetation types, as the dominant source of isoprene in the region, consistent with tropical trees being the dominant source of isoprene globally (Guenther et al. 2012).

**28. A table or figure to summarize conclusions would be helpful**

While we recognise a graphical presentation of the conclusions can be more impactful, we believe Tables 2 and 3 provide an overview of the relationships studied, and, therefore, summarise the key results. An additional figure is likely to duplicate this information, therefore we prefer not to include such a graphic.

**Technical Comments :**

**29. Line 16 : don't need "additionally" here**

While editing the introduction, the relevant sentence has been moved and the word 'additionally' is no longer used.

**30. Line 22 : "particular plants may emit different compounds more strongly or not at all" vague statement consider rewording**

We have reworded the statement (line 26-27). Specific examples of plant types and associated BVOCs are included further in the same paragraph.

While BVOCs are associated with a wide range of vegetation, particular plant species or functional types emit different amounts of specific BVOCs.

**31. Line 25 : "monoterpenes are particularly associated with needleleaf trees" particularly is leading, consider something like "primarily"**

We have updated the wording in line 30 to primarily.

For example, isoprene is associated with tropical broadleaf trees, while monoterpenes are *particularly*primarily associated with needleleaf trees.

**32. Line 155 : references fig 2b as forest change ... perhaps meant to go with the following sentence**

We have moved the figure reference to the following sentence (line 205).

Generally across the region, forest cover reduces substantially from 2001 to 2013 with a smaller decline thereafter (Fig. 2b). Over the same period, savanna and grassland expand from 46.7% to 50.5% (Fig. 2b) (see Supplement for separated savanna and grassland time series).

**33. Line 183: NO2 value is missing units**

The unitless value in this line is for AOD. We have split the sentence (lines 233-235) in two for clarity.

Several of the constituents also have co-occurring minima. Isoprene, methanol and AOD all have the lowest values of  $4.7 \pm 0.1 \times 10^{15}$  molecules cm-2,  $0.1 \pm 0.01$  ppbv, and  $0.11 \pm 0.004$ , respectively, around June. -, while HCHO, CO and NO2 remain more stable between December and June.

**34. Line 202: specify that you are referring to atmospheric constituents examined in the study, the statement could be interpreted as "all"**

We have added the text as follows in line 253.

The magnitudes of all atmospheric constituents **examined in this study** increase in the dry season when burned area is largest, suggestive of changes to the atmospheric chemistry due to a substantial pyrogenic source

**35. Line 204: "as outlines above" - outlined**

We have corrected the mistake.

**36. Line 336 : "forested" ...should this be forested area or forest ? and extra comma in "savanna/grassland, region,"**

We have updated the sentence starting on line 388 for clarity.

The other constituents are also elevated over the forested **area**, compared to the savanna/grassland<del>,</del> region, although the relative difference in atmospheric constituent between the savanna/grassland and forest categories varies.

**37. Check references to R/ R2 ... several are lower case**

We have double checked the references to the Spearman rank correlation coefficient, which we use the lower case r for, and the Ordinary Least Squares regression coefficient of determination, which we use the capital R2 for. We have clarified the caption for Table 2, which includes both the standard Spearman rank correlation coefficient and that value squared (hence r2), as follows:

Spearman rank correlation coefficients r (with the Spearman rank correlation coefficient squared:  $r^2$ , for comparison with Table 3 in brackets).

**Reviewer 2**

Specific comments:

**38. Metop-B was launched in 2013, this means that the IASI data you got from 2008 until 2012 are from Metop-A?**

Thanks for pointing out this inconsistency. We have added text in section 2.2.2 (lines 129 onwards) and Table 1 (see reviewer 1 comment 6) to acknowledge the data comes from both Metop-A and Metop-B.

The total column methanol data comes from the Infrared Atmospheric Sounding Interferometers (IASI) on board the MetOp-A and MetOp-B satellites. MetOp-B is one of threeThese Eumetsat MetOp satellites have/had (Metop-A was deorbited in 2021) sun-synchronous polar orbits at an altitude of around 817 km and local overpass times of 9:30 and 21:30.

39. It is better to add a data availability section at the end of the article, (after the code availability for example), in which you list the data you used with the corresponding links or sources where we can download them, if possible.

Thank you for this suggestion. We have expanded the code availability section to include both code and data (lines 533-537).

The isoprene data is available on request from Kelley Wells and others at the University of Minnesota. The monthly methanol data used in this study is available on Zenodo (https://doi.org/10.5281/zenodo.11472103). The level 2 HCHO data is available for download from

https://disc.gsfc.nasa.gov/datasets/OMHCHO\_003/summary. The Mopitt CO data products and the MOD08\_M3 product containing the AOD data are available for download through the Earthdata portal (https://www.earthdata.nasa.gov/). OMI NO2 data is available for download from www.temis.nl.

**40. In the caption of Fig. 5, try to be consistent with the caption of Fig. 4 when mentioning the shaded area in grey.**

For consistency with Fig. 4, we have updated the captions of Fig. 5 and Fig. 6 to:

(...). Areas over 1000 m a.s.l. based on the GMTED2010 digital elevation model (Danielson et al., 2011) have been masked on all panels and are not included in further analysis.

**41. It would help if the statistical methods are explained in more detail, with equations for instance.**

Reviewer 1 also requested this. We copy our response to Reviewer 1 (comment 17) here for completeness.

We have added the Spearman rank correlation coefficient equation (line 190-192):

$$r = 1 - \frac{6\sum_{i=1}^{n} d_i^2}{n(n^2 - 1)},$$

*"where r is the Spearman rank correlation coefficient, n is the sample size and di represents the difference between the ranks of the ith values from each sample",*

as well as the following brief descriptions of the OLS and Theil-Sen regression methods in lines 193-196.

The OLS regression aims to minimise the sum of the squared differences between the observations and the model (residual sum of squares) when fitting a linear model. The Theil-Sen regression is more robust to outliers than OLS regression, as it fits a slope and intercept based on the spatial median of parameters calculated on subpopulations of the data.

**42. Add more results in the abstract, to highlight the main findings of the study. The results mentioned in the conclusions for instance, they can be rewritten in a more concise way in the abstract.**

Thanks for pointing out this omission. Reviewer 1 had a similar request. We copy our response to Reviewer 1 (comment 1) here for completeness.

We have updated the abstract to include a more explicit list of all the metrics examined. We further include the methanol results and expand on the AOD and CO results (also relating to comment 42 from reviewer 2).

Revised abstract (lines 1 to 15):

Land surface changes can have substantial impacts on the interactions between the biosphere-atmosphere interactions. In South America, rainforests abundantly emit biogenic volatile organic compounds (BVOCs), which coupled with pyrogenic emissions from deforestation fires, can have substantial impacts on regional air quality. We use novel and long-term satellite records to characterise the impacts of biogenic and pyrogenic emissions on atmospheric composition for the period 2001 to 2019 in the southern Amazon, a region of substantial deforestation. These records include those of five trace gases: isoprene ( $C_5H_8$ ), formaldehyde (HCHO), methanol ( $CH_3OH$ ), carbon monoxide (CO), and nitrogen dioxide ( $NO_2$ ); aerosol optical depth (AOD); vegetation (land cover and leaf area index); and burned area. We find that The seasonal cycle for all of the atmospheric constituents peaks in the dry

season (August-October) and year-to-year variability in carbon monoxide-CO, formaldehyde- HCHO, nitrogen dioxide NO2, and AOD is strongly linked to burned area. We find a robust relationship between broadleaf forest cover and total column isoprene  $C_5H_8$  ( $R^2$ =0.59), while burned area exhibits an approximate 5th root power law relationship with tropospheric column NO2 ( $R^2$ =0.32), both in the dry season. Vegetation and burned area together show a relationship with HCHO ( $R^2$ =0.23). Wet season AOD and CO follow the forest cover distribution. The land surface variables are very weakly correlated with  $CH_3OH$ , suggesting other factors drive its spatial distribution. Overall, we provide a detailed observational quantification of biospheric process influences on southern Amazon regional atmospheric composition, which in future studies can be used to help constrain the underpinning processes in Earth System Models.

Minor/technical comments:

**43. Line 14-15: put a comma before and after these words "such as deforestation"**

The relevant statement has been removed while editing the introduction.

**44. Line 18: Maybe cite some of the "particular development stages"**

We have added three examples of plant development stages (lines 21/22).

BVOCs are emitted during photosynthesis and particular plant development stages, e.g. leaf maturation, flowering or senescence, or as a response to stresses on plants, such as droughts and insect infestations (Loreto and Fares, 2013).

**45. Line 21: the sentence seems like it is chopped "uncertainties in its magnitude remain.". Re-write or continue the sentence.**

Reviewer 1 also requested this. We copy our response to Reviewer 1 (comment 11) here for completeness.

We have replaced "highlighting that substantial uncertainties in its magnitude remain" with "This large range is predominantly driven by uncertainties in the emission rates from different plant functional types (Szopa et al., 2021)" on lines 24-25.

46. Line 26: for every atmospheric species you write the chemical formula, you add here for isoprene (C5H8) for instance. Like you do for methanol.

The chemical formula for isoprene is provided in line 23, when we first mention the trace gas. We have also added the formula for monoterpenes on line 30.

**47. Line 56: you wrote particular matter instead of particulate matter, also you can add (PM10 and/or PM5)**

We have corrected the error and included the acronym PM in the introduction (lines 71 and 74).

**48. Line 105: You mention that the data product has been quality assessed (Fu et al., 2019; Wells et al. 2020, 2022). Can you add a sentence to tell us the result of these assessments?**

Reviewer 1 also requested this. We copy the relevant parts of our response to Reviewer 1 (comments 8 and 16) here for completeness.

Line 127: *the retrieved isoprene column amounts differ by 20% to 50% compared to ground-based column measurements*

We have added the statement "through comparison against ground-based isoprene column measurements in the Amazon" to the sentence on line 126.

**49. Line 108: you add here the altitude of the Metop-B satellite, just like you did for OMI, EOS Aura.**

We have added the Metop altitude in line 131 (see comment 37).

**50. Line 110: add (EOS) after Earth Observing System.**

We have added the acronym.

**51. Line 178: change -average to average without the "-" in the beginning.**

We have removed the '-'.

52. Line 255: mention the lifetimes of other species when you compare the lifetime of CO to them. You do this later in the article, but it is helpful to have the lifetimes of other species mentioned earlier.

Thank you for suggesting this. Reviewer 1 had a similar suggestion (comment 6). We have added a 'lifetime' column to Table 1., which provides a broad classification of the lifetimes of the trace gases and secondary organic aerosols.

Table 1. Summary of satellite datasets analysed in the study. All datasets were analysed on  $1^{\circ} \times 1^{\circ}$  spatial resolution. Lifetimes are taken from Jacob (1999); Holloway et al. (2000); Pacifico et al. (2009); Hodzic et al. (2016); Wells et al. (2020); Bates et al. (2021); Pommier (2023)

| Variable                    | Source                | Temporal Period | Period of Data | Lifetime        |
|-----------------------------|-----------------------|-----------------|----------------|-----------------|
| Land Cover                  | MODIS, Terra and Aqua | Annual          | 2001-2020      | NA              |
| Leaf Area Index (LAI)       | MODIS, Terra          | Monthly/Annual  | 2001-2020      | NA              |
| Burned area                 | GFED4.1s              | Monthly         | 2001-2016      | NA              |
| Isoprene                    | CrIS, Suomi-NPP       | Monthly         | 2012-2020      | <1 day          |
| Methanol                    | IASI, Metop-A/B       | Monthly         | 2008-2018      | Days-Months     |
| Formaldehyde (HCHO)         | OMI, EOS Aura         | Monthly         | 2005-2018      | <1 day          |
| Carbon Monoxide (CO)        | Mopitt, Terra         | Monthly         | 2001-2019      | Months          |
| Nitrogen Dioxide (NO2)      | OMI, EOS Aura         | Monthly         | 2005-2020      | <1 day          |
| Aerosol Optical Depth (AOD) | MODIS, Terra          | Monthly         | 2000-2019      | SOA: Days-Weeks |

**53. Line 389: consider moving Fig. 9 and Fig. 10 closer to the text where you refer to them. Especially Fig. 10. Or maybe, you can merge them into one figure so it is easier for the reader to read the text, then look at the plots.**

We appreciate the feedback and recognise that the current format places the figures far from where they are discussed. We believe the exact position of the plots will change if/when the manuscript is formatted in the ACP journal article standard, as opposed to following the manuscript format as it is currently. We have prepared figures 9-11 as 1 column figures, which will allow for more flexibility in terms of figure placement in the final 2 column format, and we will check the placement of the figures during the editorial stages.